# Calculating metalation in cells reveals CobW acquires Co$^{II}$ for vitamin B$_{12}$ biosynthesis while related proteins prefer Zn$^{II}$

Tessa R. Young [1,2 ✉], Maria Alessandra Martini [1,3], Andrew W. Foster [1,2], Arthur Glasfeld[1,2,4], Deenah Osman [1,2], Richard J. Morton [5], Evelyne Deery [6], Martin J. Warren [6,7] & Nigel J. Robinson [1,2 ✉]

Protein metal-occupancy (metalation) in vivo has been elusive. To address this challenge, the available free energies of metals have recently been determined from the responses of metal sensors. Here, we use these free energy values to develop a metalation-calculator which accounts for inter-metal competition and changing metal-availabilities inside cells. We use the calculator to understand the function and mechanism of GTPase CobW, a predicted Co$^{II}$-chaperone for vitamin B$_{12}$. Upon binding nucleotide (GTP) and Mg$^{II}$, CobW assembles a high-affinity site that can obtain Co$^{II}$ or Zn$^{II}$ from the intracellular milieu. In idealised cells with sensors at the mid-points of their responses, competition within the cytosol enables Co$^{II}$ to outcompete Zn$^{II}$ for binding CobW. Thus, Co$^{II}$ is the cognate metal. However, after growth in different [Co$^{II}$], Co$^{II}$-occupancy ranges from 10 to 97% which matches CobW-dependent B$_{12}$ synthesis. The calculator also reveals that related GTPases with comparable Zn$^{II}$ affinities to CobW, preferentially acquire Zn$^{II}$ due to their relatively weaker Co$^{II}$ affinities. The calculator is made available here for use with other proteins.

$^{1}$ Department of Biosciences, Durham University, Durham, UK. $^{2}$ Department of Chemistry, Durham University, Durham, UK. $^{3}$ Max Planck Institute for Chemical Energy Conversion, Mülheim an der Ruhr, Germany. $^{4}$ Chemistry Department, Division of Mathematical and Natural Sciences, Reed College, Portland, OR, USA. $^{5}$ Department of Mathematics, Physics, and Electrical Engineering, Northumbria University, Newcastle-upon-Tyne, UK. $^{6}$ School of Biosciences, University of Kent, Canterbury, Kent, UK. $^{7}$ Quadram Institute Bioscience, Norwich Research Park, Norfolk, UK. ✉email: tessa.r.young@durham.ac.uk; nigel.robinson@durham.ac.uk

Paradoxically, in vitro, most metalloproteins prefer to bind incorrect metals[1,2]. A non-cognate metal may bind more tightly to the native site or bind by using a subset of the native ligands, by recruiting additional ligand(s) and/or by distorting the geometry of a binding site. Some enzymes are cambialistic and can function with alternative metals[3], but more commonly a non-cognate metal inactivates an enzyme[4,5]. Correct metalation occurs in vivo because cells control the availability of metals to nascent proteins[1,6–8]. For example, specialised delivery proteins support metal acquisition by about a third of metalloproteins, (which in turn represent about a third of all proteins and about a half of all enzymes)[1,8]. However, metal delivery proteins do not ultimately solve the challenge of metalation because now the correct metal must somehow partition onto the delivery protein.

For metalloproteins generally, there is a need to relate metal binding to the intracellular availability of metals. Our recent work provides the basis for such contextualisation[9]. Cells are thought to assist protein metalation by maintaining availabilities to the opposite of the Irving-Williams series with weaker binding metals such as $Mg^{II}$, $Mn^{II}$ and $Fe^{II}$ highly available and tighter binding metals such as $Ni^{II}$, $Zn^{II}$ and $Cu^{I}$ at low availabilities[10–12]. We have demonstrated this to be correct by determining the sensitivities of the DNA-binding metal-sensing transcriptional regulators of Salmonella enterica serovar Typhimurium (hereafter Salmonella)[9]. The sensors trigger expression of genes whose products, for example, import metals that are deficient or export those in excess[6,13]. A collection of thermodynamic parameters were measured for each sensor and used to calculate the (dynamic range of) buffered intracellular metal concentrations to which each sensor is finely tuned to switch gene expression[9,14]. For the more competitive metals, detection is so sensitive as to suggest that there is no hydrated metal in the cell[9,10]. Instead, rapid associative metal-exchange can occur between labile ligands in the crowded cytosol and the binding sites of metalloproteins, making it unhelpful to express metal availabilities as concentrations of the (largely irrelevant and negligible) hydrated species: thus, the chemical potentials of the bound available metals were expressed as free energies $\Delta G$[9]. It is hypothesised that metal-delivery proteins acquire their metals from these exchangeable, buffered pools. By reference to available $\Delta G$ values and by assuming an idealised cell in which the sensors are at the mid-points of their dynamic ranges, the correct metal ($Co^{II}$) was previously predicted to partition to the known chelatase of the anaerobic cobalamin biosynthetic pathway, CbiK[9]. There is a need to build upon this approach to account for (1) multiple competing metals and (2) non-idealised (conditional) cell cultures in order to understand the actions of putative metal delivery proteins (such as CobW and related GTPases) and to simplify such calculations for general use.

The G3E GTPase superfamily contains two branches of delivery proteins involved in the assembly of $Ni^{II}$ centres (HypB, UreG), one for handling the cobalamin cofactor (MeaB), plus a fourth family, COG0523, investigated here[15,16]. Though ubiquitous, from bacteria to plants and humans, members of COG0523 have been persistently enigmatic[16]. Gene context and informatics have linked subgroups of the COG0523 family to at least three different metals: these include Nha3 associated with $Fe^{III}$-requiring nitrile hydratases[17–19], various subgroups (including YeiR, ZigA and ZagA) implicated in $Zn^{II}$ metallostasis[16,20–24], and CobW associated with the aerobic biosynthesis of cobalamin (vitamin $B_{12}$) and hence $Co^{II}$ (ref. [25]). Metal insertion into the preformed corrin ring in the aerobic pathway for vitamin $B_{12}$ biosynthesis appears to be irreversible[26,27], highlighting the importance of $Co^{II}$ specificity at this step. Disruption of cobW impairs $B_{12}$ biosynthesis[25], and a role in $Co^{II}$ delivery has been

suggested[28], but not established. The roles of YeiR and YjiA (two homologues of CobW in Salmonella) are undefined, albeit $Zn^{II}$ has been predicted for YeiR[16,20], and $Co^{II}$, $Ni^{II}$ and $Zn^{II}$ shown to bind Escherichia coli (E. coli) YjiA in vitro[21]. The impact of GTP binding on metal binding remains to be tested for COG0523 GTPases.

Vitamin $B_{12}$ is an essential nutrient for human health but is neither made nor required by plants[29]. Prokaryotes present in the ruminant microbiome produce $B_{12}$ and hence meat and dairy products provide a dietary source[30]. Vitamin $B_{12}$ supplements are recommended for those on a vegan diet and its biomanufacture is in demand[31]. E. coli has significant advantages over currently employed production strains[32]. Native E. coli does not make vitamin $B_{12}$ but strains containing functional $B_{12}$ pathways have been created: the most promising of these use genes of the aerobic pathway, primarily from Rhodobacter capsulatus, and produce high levels of metal-free corrinoids[33–35]. In R. capsulatus, $Co^{II}$ is inserted into the corrin ring by a cobalt chelatase ATPase (CobNST)[36], putatively via CobW[28]. An understanding of $Co^{II}$-availability inside engineered E. coli strains (referred to hereafter as E. coli*) is required to optimise $Co^{II}$ supply for the $B_{12}$ pathway, with relevance to biomanufacturing. High $B_{12}$ production coupled with similarity between the metal sensors of E. coli and Salmonella also make this system tractable for testing metalation in vivo: the metal sensors of Salmonella having been thermodynamically characterised[9].

Here we calculate intracellular metalation to discover which metals partition onto three proposed metal delivery proteins (CobW, YeiR and YjiA). This work makes it widely possible to quantify metal occupancy of proteins and other molecules in vivo based on thermodynamic parameters. The cognate metals of proteins can thus be identified where this was uncertain, and the contributions of additional mechanisms that enable metalation (such as molecular interactions or bespoke growth conditions) exposed. We determine metal affinities of CobW, YeiR and YjiA, and calculate their in vivo metal occupancies (in Salmonella and closely related species), establishing that CobW cannot acquire $Co^{II}$ from the intracellular milieu in the absence of $Mg^{II}$GTP and revealing $Zn^{II}$ as the preferred metal for $Mg^{II}$GTP-YeiR and $Mg^{II}$GTP-YjiA. Predictions of $Co^{II}$ occupancy of $Mg^{II}$GTP-CobW in $Co^{II}$-supplemented media are reflected in CobW-dependent production of $B_{12}$ in E. coli*, establishing the function of CobW in $Co^{II}$-supply for $B_{12}$ and further validating an easy-to-use metalation calculator.

## Results

**Guanine nucleotides create two metal sites in CobW.** The first objective was to measure the $Co^{II}$ affinities of the form of CobW that acquires metal inside a cell. A modelled structure of CobW (Fig. 1a) showed hypothetical nucleotide-binding sequences adjacent to a putative metal-binding motif, CxCC, and both of these features are conserved in the COG0523 subfamily[15,16]. To assess the effect of nucleotides on metal binding, CobW was overexpressed and purified (Fig. 1b and Supplementary Fig. 1). The protein mass determined by electrospray ionisation mass spectrometry (ESI-MS) (37,071 Da; Fig. 1c) is consistent with that expected for CobW after cleavage of the N-terminal methionine (37,072.6 Da).

$Co^{II}$-titration of CobW alone (26.1 µM) produced a non-linear increase in absorbance at 315 nm (Fig. 1d) but gel-filtration of a mixture of CobW (10 µM) and $Co^{II}$ (30 µM) resulted in their complete separation (Fig. 1e). Taken together, these results suggest only weak interactions between $Co^{II}$ and CobW in the absence of cofactors. In the presence of excess GMPPNP (60 µM), a less readily hydrolysed analogue of GTP (Fig. 1f), $Co^{II}$-titration

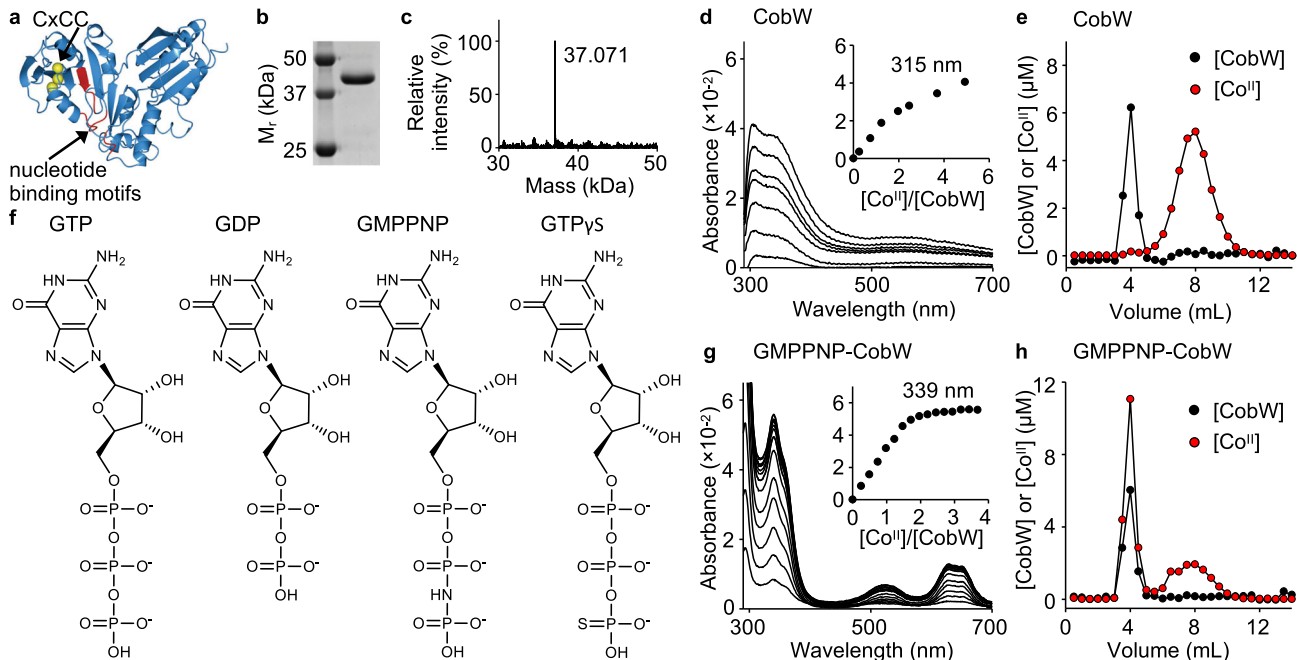

**Fig. 1 Co$^{II}$ binding to CobW is enhanced by guanine nucleotides. a** Homology model of CobW (generated with SWISS-MODEL[67] using *E. coli* YjiA PDB entry 1NIJ[68] as template; image generated using CCP4 Molecular Graphics software) showing sulfur atoms from conserved CxCC motif (in yellow) and nucleotide-binding (GxxGxGKT, hhhExxG, SKxD*) motifs[15,16] (in red). *Ordinarily NKxD but [ST]KxD observed in some COG0523 proteins[15]. **b** Purified CobW analysed by SDS-PAGE (full image in Supplementary Fig. 1; $n = 1$ under these conditions). **c** ESI-MS analysis (de-convoluted spectra) of purified CobW. **d** Representative ($n = 2$) apo-subtracted spectra of Co$^{II}$-titrated CobW (26.1 μM); feature at 315 nm (inset) shows a non-linear increase. **e** Representative ($n = 2$) elution profile following gel-filtration of a mixture of CobW (10 μM) and Co$^{II}$ (30 μM) showing no co-migration of metal (red) with protein (black). Fractions were analysed for protein by Bradford assay and for metal by ICP-MS. **f** Structures of nucleotides used in this work (generated using ChemDraw software). **g** As in **d** for a mixture of CobW (24 μM) and GMPPNP (60 μM); feature at 339 nm (inset) showing a linear increase saturating at 2:1 ratio Co$^{II}$:CobW ($n = 2$). **h** As in **e** for a mixture of CobW (10 μM), Co$^{II}$ (30 μM) and GMPPNP (30 μM) showing co-migration of 1.8 equivalents Co$^{II}$ per CobW (mean value from peak integration, $n = 2$ independent experiments). Data replicates are shown in Supplementary Fig. 1.

of CobW (24 μM) produced an absorbance feature at 339 nm characteristic of ligand-to-metal charge transfer with an extinction coefficient ($\varepsilon$ ~2800 M$^{-1}$ cm$^{-1}$) indicative of coordination by three cysteine side-chains[37] (Fig. 1g). Visible absorbance features (500–700 nm, $\varepsilon$ ~300–700 M$^{-1}$ cm$^{-1}$) are characteristic of $d$–$d$ transitions, diagnostic of tetrahedral Co$^{II}$-coordination geometry (Fig. 1g and Supplementary Fig. 2). Equivalent experiments performed with GTP and an alternate stable analogue, GTPγS, generated indistinguishable spectra (Supplementary Fig. 3a, b). These absorbance features increased linearly, saturating at 2:1 ratio Co$^{II}$:CobW, and gel filtration of a mixture of CobW (10 μM) and Co$^{II}$ (30 μM) pre-incubated with GMPPNP (30 μM) resulted in co-migration of ~2 equivalents Co$^{II}$ per protein monomer (Fig. 1h). These data show that binding of guanine nucleotides to CobW promotes tight coordination of two metal ions.

**Cellular [Mg$^{II}$] generates one Co$^{II}$ site in nucleotide-bound CobW.** The uniform absorbance increase observed across both metal-binding events in Fig. 1g could be explained by either the presence of two sequentially filled sites with identical spectroscopic features or two spectrally distinct sites being filled in a pairwise manner. Competition between GMPPNP-CobW and ethylene glycol tetraacetic acid (EGTA) or nitrilotriacetic acid (NTA) for Co$^{II}$ produced sigmoidal binding isotherms indicating positive cooperativity ($K_{D2} < K_{D1}$) between the two metal sites (Fig. 2a and Supplementary Fig. 3c, d). Such cooperativity will result in pairwise filling of the two metal sites. Given that GTPases typically bind nucleotides in complex with Mg$^{II}$, we hypothesised that the cognate metal for the first (weak-affinity)

site is Mg$^{II}$, and that Mg$^{II}$ binding triggers assembly of the second (tight-affinity) metal site in GMPPNP-CobW. Co$^{II}$ titration of CobW (20 μM) with GMPPNP (60 μM) and Mg$^{II}$ (2.7 mM, i.e. available idealised intracellular concentration, [Mg$^{II}$]$_{cell}$[9,12]) produced identical spectra to that observed without Mg$^{II}$ but saturating at 1:1 ratio Co$^{II}$:CobW (Fig. 2b and Supplementary Fig. 3e). Equivalent experiments performed with GTP and GTPγS also revealed 1:1 Co$^{II}$:CobW stoichiometry in the presence of [Mg$^{II}$]$_{cell}$ (Supplementary Fig. 3f, g). Thus, binding of Mg$^{II}$ and guanine nucleotides preassembles one distinct Co$^{II}$ site in CobW. Occupancy of the first site by Mg$^{II}$ was spectroscopically silent in these experiments. The features at 339 nm and at 500–700 nm therefore correspond exclusively to a distinct tetrahedral Co$^{II}$ site and the coordinating sulfhydryl side-chains likely derive (at least in part) from the CxCC motif adjacent to the nucleotide-binding site.

Due to the tight coordination of Co$^{II}$ to nucleotide-bound forms of CobW (i.e. no measurable dissociation at the micromolar-range protein concentrations required for detection), it was necessary to employ competition assays, whereby Co$^{II}$ is partitioned between the protein and a ligand of well-matched and defined Co$^{II}$ affinity, for reliable quantification of metal-binding affinities[38]. Competition between Mg$^{II}$GMPPNP-CobW and EGTA for Co$^{II}$ yielded a binding isotherm consistent with 1:1 stoichiometry for both Co$^{II}$:protein and Co$^{II}$:EGTA, and enabled $K_{Co(II)}$ of 2.7 ($\pm$0.4) $\times$ 10$^{-9}$ M for Mg$^{II}$GMPPNP-CobW to be determined (Fig. 2a, Supplementary Fig. 4a, b and Supplementary Tables 1, 2). Mg$^{II}$ had negligible impact on the conditional affinity of EGTA for Co$^{II}$ at the concentrations used here (Supplementary Table 3): For this reason, Mg$^{II}$ was not incorporated into curve-fitting models. Competition with EGTA

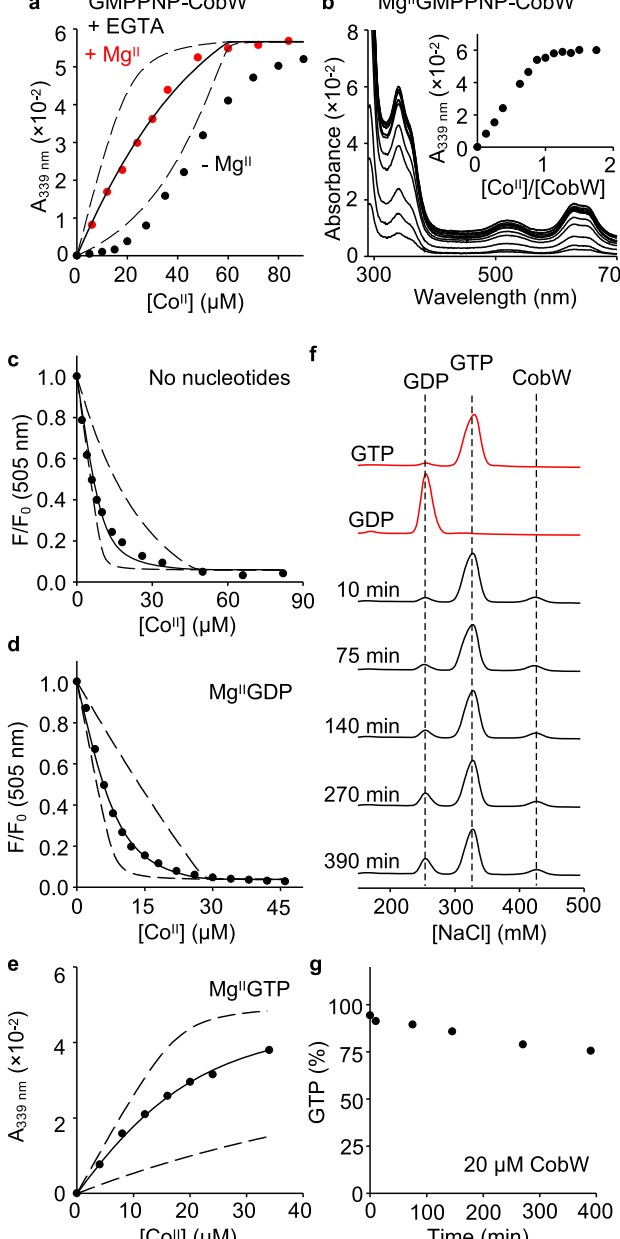

**Fig. 2 Mg$^{II}$ and the γ-phosphate group of GTP are necessary for high affinity Co$^{II}$ binding. a** Absorbance (relative to Co$^{II}$-free solution) of Co$^{II}$-titrated CobW (20 μM) with GMPPNP (60 μM) in competition with EGTA (40 μM); titrations in the absence (black) or presence (red) of Mg$^{II}$ (2.7 mM, i.e. concentration in a bacterium[9,12]). Data shown are representative of $n = 3$ independent experiments (with varying [competitor] and/or identity; see Supplementary Figs. 3c, d and 4a, b). **b** Absorbance (relative to Co$^{II}$-free solution) of Co$^{II}$-titrated CobW (20 μM) with GMPPNP (60 μM) and Mg$^{II}$ (2.7 mM) in the absence of competing ligand; feature at 339 nm (inset) showing linear increase saturating at 1:1 ratio Co$^{II}$:CobW ($n = 2$; see Supplementary Fig. 3e). **c–e** Representative $K_{Co(II)}$ quantification for CobW in the absence or presence of nucleotides ($n = 3$ independent experiments, details in Supplementary Fig. 4 and Supplementary Table 1). **c** Fluorescence quenching of Co$^{II}$-titrated fura-2 (10 μM) in competition with CobW alone (37 μM). **d** Fluorescence quenching of Co$^{II}$-titrated fura-2 (8.1 μM) in competition with CobW (20 μM) with Mg$^{II}$ (2.7 mM) and GDP (200 μM). **e** Absorbance (relative to Co$^{II}$-free solution) of Co$^{II}$-titrated CobW (18 μM) in competition with EGTA (2.0 mM) with Mg$^{II}$ (2.7 mM) and GTP (200 μM). Solid traces in **a**, **c**, **d**, **e** show curve fits of experimental data to a model where CobW binds one molar equivalent Co$^{II}$ per protein monomer. Dashed lines show simulated responses for $K_{Co(II)}$ tenfold tighter or weaker than the fitted value. **f** Analysis of GTP hydrolysis by anion-exchange chromatography. Control samples of GTP and GDP elute as distinct peaks (red traces) measured by absorbance at 254 nm. Black traces show the extent of hydrolysis of GTP (200 μM) incubated with CobW (20 μM), Mg$^{II}$ (2.7 mM) and Co$^{II}$ (18 μM) over time. **g** Analysis of data from **f** showing % GTP remaining over time. After 390 min incubation, nucleotides remain primarily (>75 %) unhydrolysed. Equivalent data using 4:1 ratio GTP:CobW is shown in Supplementary Fig. 6.

revealed a Co$^{II}$ affinity for Mg$^{II}$GTPγS-CobW ($K_{Co(II)} = 1.7$ ($\pm 0.8$) $\times 10^{-10}$ M; Supplementary Fig. 4c–e and Supplementary Tables 1, 2), that was more than tenfold tighter than Mg$^{II}$GMPPNP-CobW, establishing that the nature of the bound nucleotide exerts an effect on metal binding to CobW.

**Co$^{II}$ binds CobW 1000-fold tighter with GTP than GDP.** Observed variation in Co$^{II}$ affinities of CobW in association with Mg$^{II}$GTPγS versus Mg$^{II}$GMPPNP prompted us to assess the Co$^{II}$ affinities of all three anticipated biological species: nucleotide-free CobW, Mg$^{II}$GTP-CobW and Mg$^{II}$GDP-CobW. Co$^{II}$ affinities of CobW and Mg$^{II}$GDP-CobW were determined via competition with the probe ligand fura-2 (Fig. 2c, d, Supplementary Fig. 4f–i and Supplementary Tables 1, 2), which undergoes fluorescence quenching upon Co$^{II}$ binding[39]. Fura-2 is too weak to compete effectively with Mg$^{II}$GTP-CobW (Supplementary Fig. 4j), but high concentrations of EGTA or NTA imposed sufficient competition to enable $K_{Co(II)}$ of 3.0 ($\pm 0.8$) $\times 10^{-11}$ M to be determined (Fig. 2e, Supplementary Fig. 4k–m and Supplementary Tables 1, 2). GTP

concentration was not a limiting factor in these affinity measurements (Supplementary Fig. 5). Under identical conditions used for affinity measurements, we confirmed that CobW-catalysed GTP hydrolysis is sufficiently slow such that nucleotides remain predominantly unhydrolysed over the duration of metal-binding experiments (Fig. 2f, g and Supplementary Fig. 6). Mg$^{II}$GDP-CobW, despite displaying identical absorbance features indicating the persistence of the cysteine-rich tetrahedral site (Supplementary Fig. 7), has a Co$^{II}$ affinity more than 1000-fold weaker than Mg$^{II}$GTP-CobW and only marginally tighter than unbound CobW which lacks this site altogether (Supplementary Table 2). GTP also confers higher Co$^{II}$ affinity than either of the tested non-hydrolysable analogues in which the γ-phosphates have been modified (Fig. 1f and Supplementary Table 2). Thus, the presence of an intact nucleotide γ-phosphate is a prerequisite for high-affinity Co$^{II}$ binding.

**Cu$^{I}$ and Zn$^{II}$ bind Mg$^{II}$GTP-CobW more tightly than Co$^{II}$.** In view of the challenges associated with correct metal–protein speciation, we sought to determine Mg$^{II}$GTP-CobW affinities for other first-row transition metals (Fe$^{II}$, Ni$^{II}$, Cu$^{I}$, Zn$^{II}$). Fe$^{II}$-titration into a mixture of Mg$^{II}$GTP-CobW (50 μM) and probe ligand 4-(2-thiazolylazo)-resorcinol (Tar) (16 μM) showed Fe$^{II}$ being withheld by Tar which revealed a limiting affinity ($K_{Fe(II)} > 10^{-6}$ M) for Mg$^{II}$GTP-CobW (Fig. 3a, Supplementary Fig. 8 and Supplementary Tables 1, 2). Competition between Mg$^{II}$GTP-CobW (10 μM) and mag-fura-2 (Mf2; 20 μM) for Ni$^{II}$ showed that Mg$^{II}$GTP-CobW has one Ni$^{II}$ site which outcompetes Mf2 ($K_{Ni(II)} < 10^{-8}$ M) in addition to two weaker sites which compete with Mf2 for Ni$^{II}$ ($K_{Ni(II)} \sim 10^{-7}$ M) and are also present in CobW alone (Supplementary Fig. 9a). Competition with Tar allowed the affinity of the tight Ni$^{II}$ site in Mg$^{II}$GTP-CobW to be determined ($K_{Ni(II)} = 9.8$ ($\pm 6.5$) $\times 10^{-10}$ M; Fig. 3b, Supplementary Fig. 9b, c and Supplementary Tables 1, 2). The conditional $\beta_2$ value

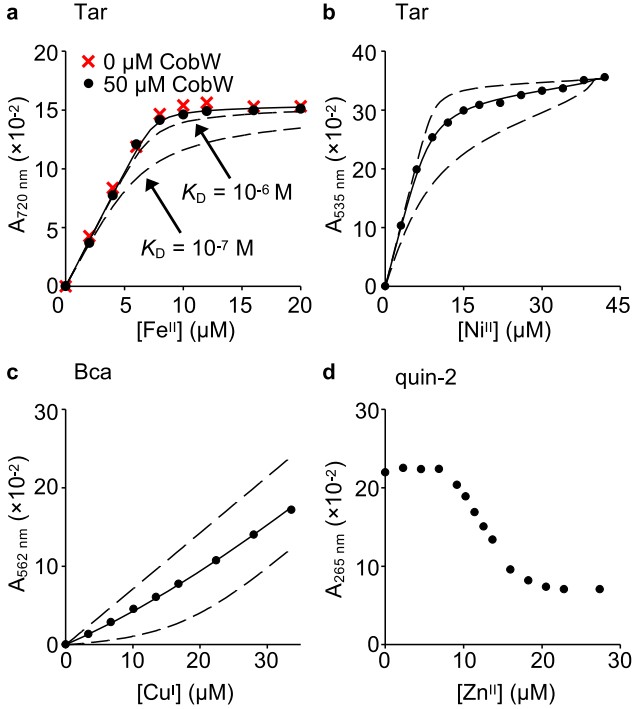

**Fig. 3 Binding of $Mg^{II}GTP$-CobW to $Fe^{II}$, $Ni^{II}$, $Cu^I$ and $Zn^{II}$. a** Absorbance upon $Fe^{II}$ titration into a mixture of Tar (16 μM), $Mg^{II}$ (2.7 mM) and GTP (500 μM) in the absence (red crosses) or presence (black circles) of CobW (50 μM). Dashed lines show simulated responses for specified $K_{Fe(II)}$ of $Mg^{II}GTP$-CobW, providing limiting $K_{Fe(II)} \geq 10^{-6}$ M. Control $Fe^{II}$ titration into a solution of Tar (16 μM) in buffer only (Supplementary Fig. 8a) confirmed that $Mg^{II}$ and GTP did not inhibit stoichiometric $Fe^{II}Tar_2$ formation. **b** Absorbance change (relative to $Ni^{II}$-free solution) of $Ni^{II}$-titrated Tar (20 μM) in competition with CobW (30 μM) in the presence of $Mg^{II}$ (2.7 mM) and GTP (300 μM). **c** Absorbance of $Cu^I$-titrated Bca (1.0 mM) in competition with CobW (20 μM) in the presence of $Mg^{II}$ (2.7 mM) and GTP (200 μM). In **a**–**c**, solid traces show curve fits of experimental data to models where CobW binds one molar equivalent of metal per protein monomer. Supplementary Table 2 contains mean ± standard deviation (SD) $K_{metal}$ values from $n = 3$ independent experiments (full details in Supplementary Figs. 8–12 and Supplementary Table 1). In **b**, **c**, dashed lines show simulated responses for $K_{metal}$ tenfold tighter or weaker than the fitted value. **d** Absorbance (relative to probe-free solution) upon titration of $Zn^{II}$ into a mixture of quin-2 (10 μM), $Mg^{II}$ (2.7 mM), GTP (100 μM) and CobW (10 μM).

$(4.3 \ (\pm 0.6) \times 10^{15} \ M^{-2})$ for $Ni^{II}Tar_2$ formation under experimental conditions (pH 7.0, 100 mM NaCl, 400 mM KCl) was independently established by competition with EGTA (Supplementary Fig. 10). Titration of $Mg^{II}GTP$-CobW (15 μM) and bathocuproine disulfonate (Bcs; 30 μM) with $Cu^I$ did not reach the expected intensity at saturating metal concentrations (Supplementary Fig. 11a) suggesting the presence of a stable ternary complex, which would preclude accurate affinity determinations[40]. An equivalent experiment with alternative $Cu^I$-probe bicinchoninic acid (Bca) showed that $Mg^{II}GTP$-CobW has two $Cu^I$ sites which outcompete Bca and at least three additional weaker $Cu^I$ sites which effectively compete with the probe (Supplementary Fig. 11b). Effective competition imposed by excess Bca enabled $K_{Cu(I)}$ of $2.4 \ (\pm 0.9) \times 10^{-16}$ M to be determined (Fig. 3c, Supplementary Figs. 11c, d, 12 and Supplementary Tables 1, 2), assuming only the tightest $Cu^I$ site can acquire metal at the limiting $Cu^I$ availabilities employed (e.g. $[Cu^I_{aq}] < 3 \times 10^{-16}$ M in Fig. 3c). $Zn^{II}$ titration into a mixture of quin-2 (10 μM) and $Mg^{II}GTP$-CobW (10 μM) revealed one high-affinity $Zn^{II}$ site in

the protein which was too tight to be quantified by using quin-2 thus showing $K_{Zn(II)} < 10^{-12}$ M (Fig. 3d).

Because of the limiting affinity of quin-2, we employed inter-metal competition, which presumably also occurs within the buffered intracellular milieu, to determine $K_{Zn(II)}$ for $Mg^{II}GTP$-CobW. $K_{Zn(II)}$ was determined, relative to the known $K_{Co(II)}$, via competition between the two metals. This approach required an excess of metal ions competing for a limited number of protein metal-sites (i.e. $[Co^{II}]_{tot} + [Zn^{II}]_{tot} > [CobW]_{tot}$), thus it was essential to include a buffering ligand, in this case NTA, to control the speciation of all $Co^{II}$ and $Zn^{II}$ in solution (i.e. $[NTA]_{tot} > [Co^{II}]_{tot} + [Zn^{II}]_{tot}$). The measured equilibrium ($K_{ex}$ in Fig. 4a) was the exchange constant for $Co^{II}/Zn^{II}$ exchange between the protein ($Mg^{II}GTP$-CobW) and buffering ligand (NTA). Equilibrium ratios of $[Co^{II}Mg^{II}GTP$-CobW]/$[Zn^{II}M-g^{II}GTP$-CobW] were determined (Fig. 4b–e and Supplementary Table 4): absorbance intensity at $A_{339 \ nm}$ reported specifically on the $Co^{II}$–protein complex and all remaining protein was $Zn^{II}$-bound (since $Mg^{II}GTP$-CobW was metal-saturated under experimental conditions; Supplementary Fig. 13). The concentrations of NTA-bound metals were determined from mass balance relationships (Eqs. (6–8) in "Methods"). Experiments were conducted at multiple relative availabilities of $Co^{II}$ and $Zn^{II}$ and reciprocally (Fig. 4b–e), with consistent results (Supplementary Table 4), to confirm reliability of measured affinities. We thus determined a tight $K_{Zn(II)}$ of $1.9 \ (\pm 0.6) \times 10^{-13}$ M for $Mg^{II}GTP$-CobW (Supplementary Table 2).

**GTP not GDP enables $Co^{II}$ acquisition by CobW in cells.** In the same manner that Fig. 3 considered competition between a ligand (Tar, Bca or quin-2) and a protein ($Mg^{II}GTP$-CobW) for metal binding in vitro, metal acquisition by proteins in vivo likewise involves competition with a surplus of cytosolic ligands that buffer metals to different availabilities[8,9,14,41,42]. Recent work has estimated the buffered availabilities of metals M (where M = $Mg^{II}$, $Mn^{II}$, $Fe^{II}$, $Co^{II}$, $Ni^{II}$, $Cu^I$, $Zn^{II}$) in a reference bacterium (*Salmonella*[9]) expressed here as free energies ($\Delta G$; Fig. 5). The intracellular available $\Delta G$ for each metal, $\Delta G_M$, is defined as the free energy required for a ligand to become 50% metalated from available and exchangeable intracellular metal (see Supplementary Note 1). Figure 5 and Supplementary Fig. 14 show the intracellular available $\Delta G_M$ values in an idealised cell (i.e. neither metal deficiency nor excess) defined as the metal availabilities at which the cognate sensors undergoes half of their transcriptional responses. Bars show the changes in available intracellular $\Delta G_M$ as sensors shift from 10–90% (Fig. 5) or 1–99% (Supplementary Fig. 14) of their dynamic ranges. The percentage occupancy of a protein, P, with metal, M, in vivo is governed by the difference between the free energy for protein metalation, $\Delta G_{MP}$, and the intracellular available $\Delta G_M$ (Eq. (1)) and can be calculated via Eq. (2) (see Supplementary Note 1):

$$\Delta\Delta G_M = \Delta G_{MP} - \Delta G_M \qquad (1)$$

$$\text{Fractional occupancy (\%)} = 100 \times \frac{[MP]}{[P]_{tot}} = 100 \times \frac{e^{\frac{-\Delta\Delta G_M}{RT}}}{1 + e^{\frac{-\Delta\Delta G_M}{RT}}} \qquad (2)$$

In an idealised cell, the $\Delta G_{Co(II)}$ for CobW and $Mg^{II}GDP$-CobW were both significantly more positive than intracellular available $\Delta G_{Co(II)}$ ($\Delta\Delta G_{Co(II)} \gg 0$; Fig. 5) resulting in negligible $Co^{II}$ occupancies of 1.0% and 2.5% for these two protein forms, respectively. Conversely, $\Delta G_{Co(II)}$ for $Mg^{II}GTP$-CobW was significantly more negative than intracellular available $\Delta G_{Co(II)}$

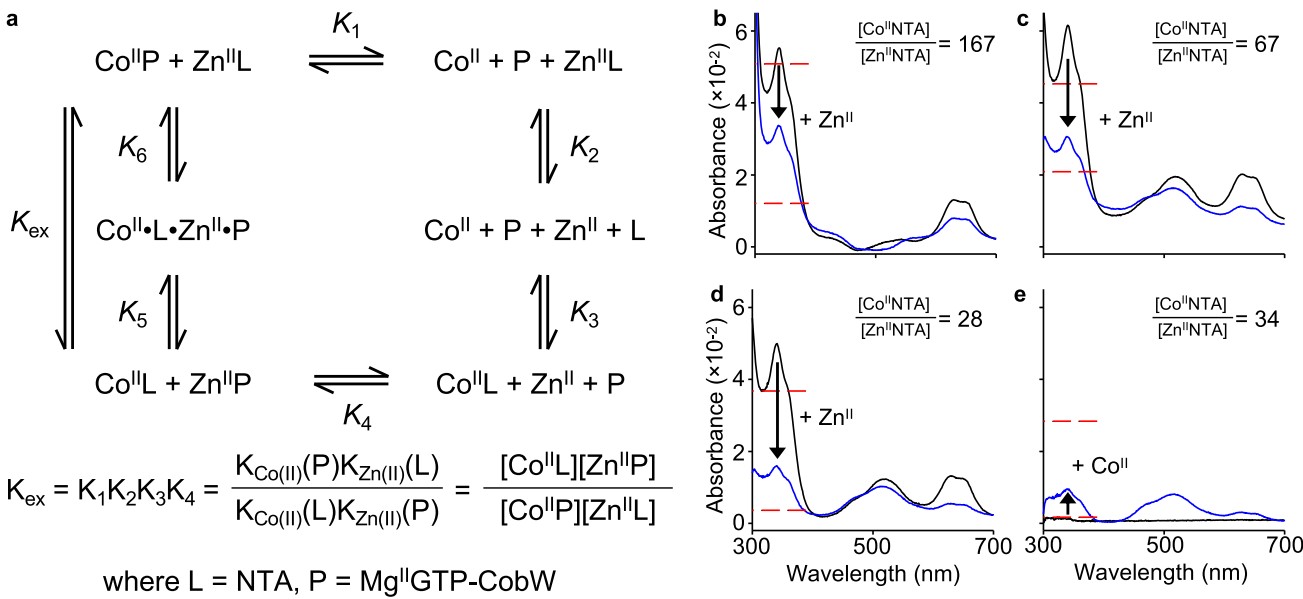

**Fig. 4 Mg$^{II}$GTP-CobW binds Zn$^{II}$ with sub-picomolar affinity. a** Representation of the equilibrium for exchange of Co$^{II}$ and Zn$^{II}$ between ligand (L = NTA) and protein (P = Mg$^{II}$GTP-CobW). **b–e** Absorbance (relative to metal-free solution) of solutions of CobW (17.9–20.4 μM), Mg$^{II}$ (2.7 mM), GTP (200 μM) and NTA (0.4–4.0 mM) upon (**b–d**) first addition of Co$^{II}$ (black trace) then Zn$^{II}$ (blue trace) or (**e**) the reverse, at equilibrium (n = 1 for each panel). The absorbance peak at 339 nm corresponds to Co$^{II}$-bound protein. An excess of ligand NTA was used to buffer both metals in each experiment: varying the ratios of ligand-bound metal ions ([Co$^{II}$NTA]/[Zn$^{II}$NTA] = 28–167) shifted the ratios of Co$^{II}$- and Zn$^{II}$-bound protein as predicted by the equilibrium exchange constant in **a**. Consistent $K_{Zn(II)}$ values for Mg$^{II}$GTP-CobW were generated at all tested conditions (Supplementary Table 4). Dashed red lines show expected $A_{339\ nm}$ peak intensities for $K_{Zn(II)}$ of Mg$^{II}$GTP-CobW tenfold tighter or weaker than calculated values.

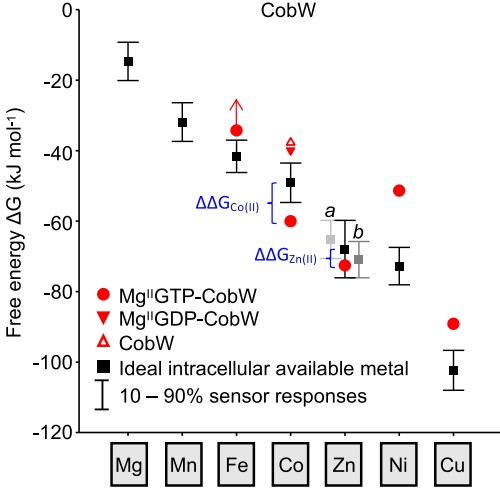

**Fig. 5 Mg$^{II}$GTP-CobW is predicted to acquire Co$^{II}$ or Zn$^{II}$ in a bacterial cell.** Free-energy change (ΔG) for metal binding to Mg$^{II}$GTP-CobW (red circles) plotted against the intracellular available free energies for metal binding in a reference bacterial cytosol (values correspond to *Salmonella*) under idealised conditions (i.e. where each metal sensor undergoes half of its transcriptional response; black squares). Intracellular available $ΔG_{Zn(II)}$ is the mean of the values determined from the two Zn$^{II}$ sensors ZntR (*a*) and Zur (*b*). Bars show the change in intracellular available ΔG as cognate sensors shifts from 10–90% of their responses. Free energy differences (ΔΔG) which favour acquisition of metals by Mg$^{II}$GTP-CobW in vivo are indicated in blue. ΔG values for Co$^{II}$ complexes of CobW alone (open red triangle) and Mg$^{II}$GDP-CobW (closed red triangle) are also shown. For Fe$^{II}$ binding to Mg$^{II}$GTP-CobW, arrow indicates limiting ΔG > −34.2 kJ mol$^{−1}$.

**Table 1 Calculated metal occupancies of COG0523 proteins in an idealised cell[a].**

| Metal | Mg$^{II}$GTP-CobW Eq. 2[b] | Mg$^{II}$GTP-CobW Eq. 4[c] | Mg$^{II}$GTP-YeiR Eq. 4[c] | Mg$^{II}$GTPγS-YjiA Eq. 4[c] |
|---|---|---|---|---|
| Mn$^{II}$ [d] | | | <0.8% | <1.8% |
| Fe$^{II}$ [d] | <4.6% | <0.1% | <3.0% | < 3.4% |
| Co$^{II}$ | 98.8% | 91.9% | 10.3% | 2.0% |
| Zn$^{II}$ | 86.2% | 6.9% | 24.4% | 22.6% |
| Ni$^{II}$ | 0.1% | 0.0% | 0.0% | 0.0% |
| Cu$^{I}$ | 0.5% | 0.0% | 0.2% | 0.1% |
| Total | 190.3% | 98.9% | 38.6% | 29.8% |

[a]Based on metal availabilities in *Salmonella* under idealised conditions (ref. [9]).
[b]Does not account for competition between different metals for the same high-affinity site in Mg$^{II}$GTP-CobW.
[c]Takes into account competition between multiple intracellular metals for the same site in each protein.
[d]Where only limiting metal–protein affinities (strongest $K_D$ limit) were determined, calculated occupancy corresponds to a maximum value (denoted by <).

($ΔΔG_{Co(II)} ≪ 0$), resulting in almost complete protein metalation (99%). Thus, CobW needs Mg$^{II}$GTP to acquire Co$^{II}$ in a cell.

**Mg$^{II}$GTP-CobW may also acquire Zn$^{II}$.** In addition to Co$^{II}$, other metals also bound to Mg$^{II}$GTP-CobW (Figs. 3 and 4). However, ΔΔG for Fe$^{II}$, Ni$^{II}$ and Cu$^{I}$ was significantly greater than zero (Eq. (1) and Fig. 5), thus preventing acquisition of these metals (Eq. (2) and Table 1). In contrast, $ΔΔG_{Zn(II)}$ was <0 with in vivo Zn$^{II}$ occupancy predicted to be 86% (Fig. 5 and Table 1). However, based on Eq. (2), the sum of metal occupancies of Mg$^{II}$GTP-CobW gave an impossible total metalation >100%

(Table 1). Since $\Delta\Delta G$ was <0 for both $Co^{II}$ and $Zn^{II}$, a more sophisticated approach needs to account for competition between multiple buffered metals in order to predict how much $Zn^{II}$ binds $Mg^{II}GTP$-CobW in vivo.

**Calculating inter-metal competition in a cell.** Figure 4 considered competition between $Co^{II}$ and $Zn^{II}$ for a single metal-binding site in a protein ($Mg^{II}GTP$-CobW) when the metals were buffered to different availabilities in vitro by an excess of NTA. This can be represented as an available $\Delta G_M$ (Supplementary Table 4). The observed $Co^{II}$ occupancy was a function of the protein's affinities for both $Co^{II}$ and $Zn^{II}$ relative to their buffered availabilities in solution (i.e. $\Delta\Delta G$ values), as described by Eq. (3) (see Supplementary Note 1).

$$\text{Fractional (\%) } Co^{II} \text{ occupancy} = 100 \times \frac{e^{\frac{-\Delta\Delta G_{Co(II)}}{RT}}}{1 + e^{\frac{-\Delta\Delta G_{Co(II)}}{RT}} + e^{\frac{-\Delta\Delta G_{Zn(II)}}{RT}}}$$

(3)

By analogy, in a cytoplasm multiple metals, each buffered to different intracellular available $\Delta G_M$, compete for a single protein-binding site. We generalised Eq. (3) to account for $n$ different metals (Eq. (4) and Supplementary Note 1).

$$\text{Fractional (\%) occupancy (with metal } M_1 \text{ of interest)} = 100 \times \frac{e^{\frac{-\Delta\Delta G_{M1}}{RT}}}{1 + \sum_{k=1}^{k=n} e^{\frac{-\Delta\Delta G_{Mk}}{RT}}}$$

(4)

Thus, we developed a metalation calculator (based on *Salmonella*, Supplementary Data 1) for determining in vivo metal occupancies of proteins, accounting for multiple inter-metal competitions plus competition from components of the intracellular milieu.

**$Mg^{II}GTP$-CobW selects $Co^{II}$ in idealised (*Salmonella*) cells.** Since $\Delta\Delta G$ was <0 for binding of both $Co^{II}$ and $Zn^{II}$ to $Mg^{II}GTP$-CobW (Fig. 5), Eq. (4) was next used to predict in vivo metalation in an idealised cell. Between the five metals considered ($Fe^{II}$, $Co^{II}$, $Ni^{II}$, $Cu^{I}$ and $Zn^{II}$), $Mg^{II}GTP$-CobW will favour $Co^{II}$ binding in a cell and calculations via Eq. (4) predicted occupancies of 92% and 7%, for $Co^{II}$ and $Zn^{II}$, respectively (Table 1). Thus, although $Mg^{II}GTP$-CobW affinities for both $Co^{II}$ and $Zn^{II}$ are tight enough to extract either metal from the cytosolic buffer, $Co^{II}$ will outcompete $Zn^{II}$, rationalising specificity but only in an intracellular context where there is competition from other cellular components.

**Related GE3 GTPase YeiR prefers $Zn^{II}$ in idealised *Salmonella*.** To test the calculator on a second protein, YeiR from *Salmonella* was overexpressed and purified (Supplementary Fig. 15) in order to determine metal affinities. In view of similarity between YeiR, ZigA[22,23] and ZagA[24], notably a deduced binding site for $Zn^{II}$-sensor Zur in the *yeiR* promoter (Supplementary Fig. 16), occupancy with $Zn^{II}$ might be predicted. Perhaps unexpectedly, $Mg^{II}GTP$-YeiR showed a similar (slightly weaker) affinity for $Zn^{II}$ relative to $Mg^{II}GTP$-CobW, the greatest difference in affinity was for $Co^{II}$ (Supplementary Figs. 17–21 and Supplementary Tables 5, 6).

$Mn^{II}$ failed to migrate through a gel filtration column with $Mg^{II}GTP$-YeiR even when the running buffer was supplemented with an additional 20 μM $MnCl_2$, revealing a $Mn^{II}$ affinity $>2 \times 10^{-4}$ M (Supplementary Figs. 17a and 18). $Mg^{II}GTP$-YeiR did not compete with Tar under conditions that imply $Fe^{II}$ affinity $\geq 1 \times 10^{-6}$ M (Supplementary Fig. 17b). The $Co^{II}$ and $Ni^{II}$ affinities of $Mg^{II}GTP$-YeiR were determined by competition with fura-2 and Mf2, respectively (Supplementary Fig. 17c, d and Supplementary Fig. 19a–d). Data were fit to a 1:1 metal-binding model giving a $Co^{II}$ affinity of 1.5 ($\pm 0.7$) $\times 10^{-8}$ M and $Ni^{II}$ affinity of 1.5 ($\pm 0.6$) $\times 10^{-7}$ M. Competition for $Cu^{I}$ between $Mg^{II}GTP$-YeiR and Bca identified a $Cu^{I}$ affinity of 4.9 ($\pm 5.1$) $\times 10^{-16}$ M (Supplementary Fig. 17e). Competition with quin-2 was used to determine $Zn^{II}$ affinities of $Mg^{II}GTP$-YeiR of 3.0 ($\pm 1.2$) $\times 10^{-12}$ M, and 4.1 ($\pm 2.7$) $\times 10^{-12}$ M for $Mg^{II}GTP\gamma S$-YeiR (Supplementary Figs. 17f, 19e–g, 20d and 21h, i). Equation (4) was used to predict in vivo metalation of $Mg^{II}GTP$-YeiR in an idealised cell. $Zn^{II}$ binding is favoured with calculated occupancies of 24% and 10%, for $Zn^{II}$ and $Co^{II}$ respectively, when sensors are at the mid-points of their dynamic ranges (Fig. 6a and Table 1), and trace amounts of zinc were detected after extensive purification (Supplementary Figs. 22 and 23).

**Related GE3 GTPase YjiA prefers $Zn^{II}$ in idealised *Salmonella*.** $Co^{II}$, $Ni^{II}$ and $Zn^{II}$ have all been shown to bind recombinant YjiA in vitro[21], and the *yjiA* promoter contains no deduced recognition sequence for Zur. To test the calculator on a third COG0523 protein, YjiA was overexpressed, purified (Supplementary Fig. 15), and its affinities for metals determined (Supplementary Figs. 24 and 25, Supplementary Tables 5 and 7).

Affinities were determined for $Mg^{II}GTP\gamma S$-YjiA to avoid nucleotide hydrolysis: this is supported by similar $Zn^{II}$ affinities being measured for $Mg^{II}GTP$-YeiR and $Mg^{II}GTP\gamma S$-YeiR

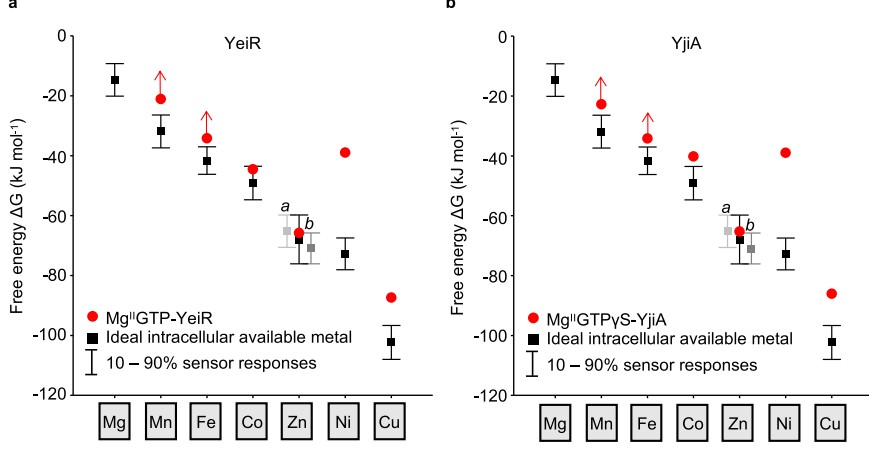

**Fig. 6 $Mg^{II}GTP$-YeiR and $Mg^{II}GTP$-YjiA preferentially acquire $Zn^{II}$. a** Free-energy change ($\Delta G$) for metal binding to $Mg^{II}GTP$-YeiR (red circles) plotted against the intracellular available free energies for metal binding (as described in Fig. 5; black squares and bars). **b** As **a** for $Mg^{II}GTP\gamma S$-YjiA (red circles). Arrows indicate where only a limiting $\Delta G$ was determined (thus $\Delta G >$ plotted value).

(Supplementary Table 5). Mf2 fully outcompeted $Mg^{II}GTP\gamma S$-YjiA for $Mn^{II}$ and simulations indicate a dissociation constant $\geq 1 \times 10^{-4}$ M (Supplementary Figs. 24a and 25a–c). $Mg^{II}GTP\gamma S$-YjiA did not compete with Tar under conditions that imply $Fe^{II}$ affinity $\geq 1 \times 10^{-6}$ M (Supplementary Figs. 24b and 25d–f). The $Co^{II}$, $Ni^{II}$ and $Cu^{I}$ affinities of $Mg^{II}GTP\gamma S$-YjiA were determined using fura-2, Mf2 and Bca, respectively, as described for YeiR, giving a $Co^{II}$ affinity of 9.1 $(\pm 2.0) \times 10^{-8}$ M, $Ni^{II}$ affinity of 1.5 $(\pm 0.3) \times 10^{-7}$ M and $Cu^{I}$ affinity of 7.6 $(\pm 1.4) \times 10^{-16}$ M (Supplementary Figs. 24c–e and 25g–l). Competition with quin-2 was used to determine a $Zn^{II}$ affinity for $Mg^{II}GTP\gamma S$-YjiA of 3.7 $(\pm 1.1) \times 10^{-12}$ M, and this was repeated with $Mg^{II}GTP$-YjiA generating a near identical value of 3.3 $(\pm 2.5) \times 10^{-12}$ M (Supplementary Figs. 24f and 25m–p). Equation (4) was used to predict in vivo metalation of $Mg^{II}GTP$-YjiA in an idealised cell. $Zn^{II}$ binding is favoured with occupancies of 23% and 2.0% for $Zn^{II}$ and $Co^{II}$, respectively (Fig. 6b and Table 1). Notably, the two $Zn^{II}$ sensors show a relatively wide dynamic range for $\Delta G_{Zn(II)}$, suggesting that $Zn^{II}$ occupancy could increase dependent upon media $[Zn^{II}]$.

**$Mg^{II}GTP$-CobW outcompetes $Mg^{II}GTP$-YeiR for $Co^{II}$.** Counterintuitively, $Mg^{II}GTP$-YeiR and $Mg^{II}GTP$-YjiA were predicted to preferentially bind $Zn^{II}$ in vivo, not due to tighter affinities for $Zn^{II}$, but rather due to their weaker $Co^{II}$ affinities relative to $Mg^{II}GTP$-CobW (Table 1, Figs. 5 and 6, and Supplementary Tables 2 and 5). To test relative $Co^{II}$ affinities, $Mg^{II}GTP$-YeiR was competed against $Mg^{II}GTP$-CobW (Fig. 7). YeiR and CobW were incubated with $Co^{II}$ in the presence of $Mg^{II}$ and GTP, then separated by anion exchange chromatography. The chromatography was also conducted with each protein separately. Individually each protein eluted bound to $Co^{II}$, but in competition $Co^{II}$ eluted almost exclusively with $Mg^{II}GTP$-CobW (Fig. 7 and Supplementary Fig. 26), confirming its tighter affinity for $Co^{II}$.

**Fine tuning $\Delta G$ for metalation in a cell.** Calculated free energies for intracellular metalation ($\Delta G_M$) in Figs. 5 and 6 are based on an assumption that cellular metal availabilities are fixed at ideal buffered concentrations where every metal sensor undergoes half of its transcriptional response (i.e. normalised fractional DNA

occupancy $\theta_D = 0.5$, see ref. [9]). In reality, cellular metal availabilities, and consequently $\theta_D$ of sensors, fluctuate conditionally (e.g. in response to addition of metals or chelators to the growth media). For example, the dynamic response range (defined as $\theta_D = 0.99$–0.01) of RcnR, the $Co^{II}$ sensor from *Salmonella*, coincides with an increase in the intracellular available $[Co^{II}]$ from $2.4 \times 10^{-11}$ to $2.7 \times 10^{-7}$ M, corresponding to an increase in intracellular available $\Delta G_{Co(II)}$ from $-60.6$ to $-37.5$ kJ mol$^{-1}$ (Fig. 8a and Supplementary Table 8).

In order to account for this variation, we developed a method to fine-tune free energy calculations under bespoke culture conditions using quantitative polymerase chain reaction (qPCR) analyses of transcripts regulated by metal sensors. Fine-tuning was performed for $Co^{II}$ in *E. coli** which has been engineered to synthesise vitamin $B_{12}$ (*E. coli* and *Salmonella* RcnR share 93% sequence identity and equivalent responses to available $Co^{II}$ were

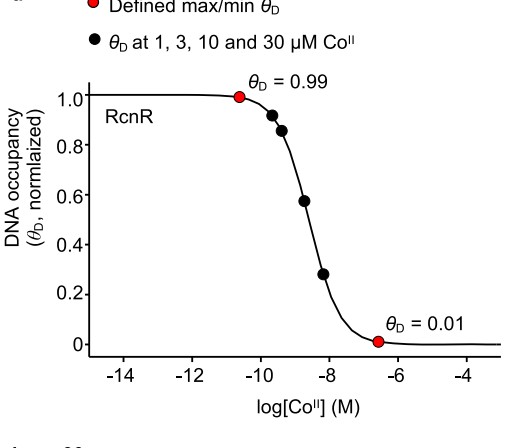

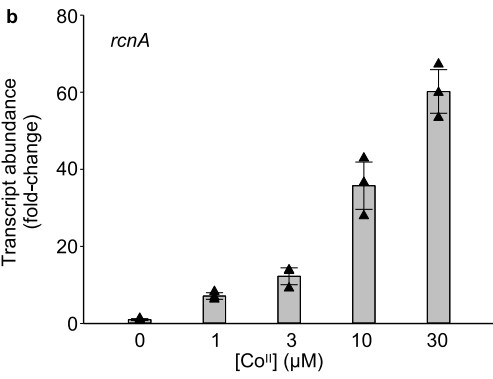

**Fig. 8 Calculations of conditional $Co^{II}$ availabilities in $B_{12}$-producing *E. coli**. a** Calculated relationship between intracellular $Co^{II}$ availability and normalised DNA occupancy ($\theta_D$) by RcnR. $\theta_D$ of 0 and 1 are the maximum and minimum calculated DNA occupancies. The dynamic range (within which RcnR responds to changing intracellular $Co^{II}$ availability) has been defined as $\theta_D$ of 0.01–0.99 (i.e. 1–99% of RcnR response). The calibrated maximum and minimum fold changes in *rcnA* transcript abundance (i.e. boundary conditions, see Supplementary Fig. 27) therefore correspond to $\theta_D$ of 0.01 and 0.99 in these calculations (red circles). $\theta_D$ for each growth condition (black circles) was calculated from the qPCR response in **b**, assuming a linear relationship between change in $\theta_D$ and change in transcript abundance (Eq. (10)). Corresponding $Co^{II}$ availabilities are listed in Supplementary Table 8. **b** Transcript abundance (relative to untreated control) of the RcnR-regulated gene *rcnA* following 1 h exposure of *E. coli** to increasing $[Co^{II}]$, measured by qPCR. Data are the mean ± SD of $n = 3$ biologically independent replicates. Triangle shapes represent individual experiments (some data points overlap, experimental values are available in Source Data files).

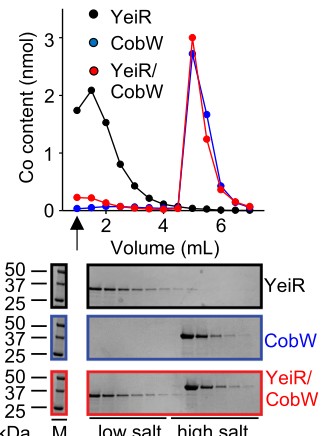

**Fig. 7 $Mg^{II}GTP$-CobW outcompetes $Mg^{II}GTP$-YeiR for $Co^{II}$.** Elution profile of YeiR (10 µM), CobW (10 µM) or both proteins following incubation with GTP (100 µM), $Mg^{II}$ (2.7 mM) and $Co^{II}$ (8 µM) resolved by differential elution from an anion exchange column. Fractions were analysed for $Co^{II}$ by ICP-MS and protein by SDS-PAGE (YeiR alone black, CobW alone blue, both proteins red; $n = 1$). Arrow denotes flow through fractions. Full gel images and SDS-PAGE analysis of flow through fractions shown in Supplementary Fig. 26.

assumed). *E. coli** cells were cultured in Luria-Bertani (LB) medium with increasing $Co^{II}$ supplementation. Abundance of the RcnR-regulated *rcnA* transcript (Fig. 8b) was used to calculate $\theta_D$ of RcnR for each condition (via Eq. (10) in "Methods") following calibration of the maximum and minimum responses (defined as $\theta_D = 0.99$ and 0.01 at low and high $[Co^{II}]$, respectively; Supplementary Fig. 27). This enabled the intracellular $Co^{II}$ availabilities, as conditional free energies, to be calculated from the RcnR $\theta_D$ for each condition (Fig. 8a and Supplementary Table 8).

**$Co^{II}$-acquisition by $Mg^{II}GTP$-CobW predicts $B_{12}$ synthesis.** Does the amount of $Co^{II}$ inserted into $B_{12}$ follow the predicted metalation of $Mg^{II}GTP$-CobW? Metal occupancies of $Mg^{II}GTP$-CobW in *E. coli** samples were recalculated (via Eq. (4)) using bespoke intracellular available free energies, $\Delta G_{Co(II)}$, for each growth condition (Fig. 8 and Supplementary Table 8). This predicted that in unsupplemented LB media the protein would be predominantly $Zn^{II}$-bound (10% $Co^{II}$ and 77% $Zn^{II}$) and that $Co^{II}$ occupancies would increase from 10% to 97% as added $[Co^{II}]$ increased from 0 to 30 µM (Fig. 9a). Since intracellular $Zn^{II}$ availability was also significant in our predictions, we confirmed that our previous estimation of $\Delta G_{Zn(II)}$ was valid in LB media (Supplementary Fig. 28). Corrin concentrations (presumed to be predominantly $B_{12}$, noting that intermediates after $Co^{II}$ insertion may also be detected, and that $Zn^{II}$ may competitively inhibit the chelatase complex but not insert into ring-contracted corrins[36]) were measured in *E. coli** strains containing or missing *cobW* (Fig. 9b and Supplementary Fig. 29), under the growth conditions for which intracellular available $\Delta G_{Co(II)}$ was defined (Supplementary Table 8). As the added $[Co^{II}]$ increased so did $B_{12}$ production in *cobW*(+), consistent with the predicted loading of $Mg^{II}GTP$-CobW with $Co^{II}$ (Fig. 9). At higher $[Co^{II}]$, CobW-independent $B_{12}$ synthesis became evident. As anticipated, total cellular cobalt increases with supplementation, and the amount of cobalt in $B_{12}$ is <10% of the total cellular cobalt (Supplementary Table 9). The number of additional atoms accumulated per cell exceeds the amount predicted if $Co^{II}$ were not buffered, noting that the internal buffered concentration at 10 µM exogenous $Co^{II}$ is 1.9 nM (Fig. 8 and Supplementary Table 8), and that only 1 atom per cell volume (approximately 1 femtolitre) equates to 1.7 nM. Most importantly, $B_{12}$ synthesis which is dependent on CobW (Fig. 9b, compare *cobW*(+) with *cobW*(−)) matches the trend in predicted metalation of $Mg^{II}GTP$-CobW (Fig. 9a).

## Discussion

Here we relate metal affinities of three putative metallochaperones to a thermodynamic framework, identifying their cognate metals which align with previous speculations[16,20,25] (Figs. 5, 6 and Table 1). We establish the connection between CobW and $Co^{II}$ and show how CobW can acquire $Co^{II}$ in a cell (Figs. 1–5 and Table 1). Free-energy calculations reveal that in an idealised cell $Co^{II}$ ions will not flow from the cellular milieu to nucleotide-free CobW ($\Delta\Delta G_{Co(II)} > 0$). Crucially, $Co^{II}$ will flow from the cellular milieu to the $Mg^{II}GTP$ form of CobW ($\Delta\Delta G_{Co(II)} < 0$) (Fig. 5, Supplementary Fig. 30a, Table 1 and Supplementary Table 2). Thus, CobW must first bind $Mg^{II}GTP$ in order to acquire $Co^{II}$ inside a cell. In contrast, the product of GTP hydrolysis, $Mg^{II}GDP$-CobW, will release $Co^{II}$ to the cellular milieu ($\Delta\Delta G_{Co(II)} > 0$) (Fig. 5, Supplementary Fig. 30b and Supplementary Table 2). Thus, the GTPase activity of CobW will facilitate $Co^{II}$ release, for example to CobNST for insertion into the corrin ring of $B_{12}$ (Fig. 2f, g and Supplementary Fig. 6). We establish that CobW enhances $B_{12}$ production when $Co^{II}$ is limiting (Fig. 9b), and Supplementary Fig. 30 illustrates the proposed

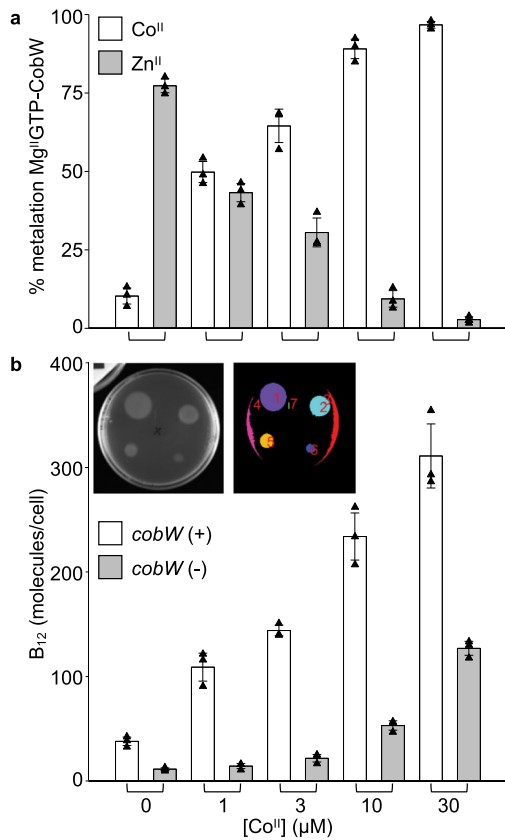

**Fig. 9 $B_{12}$ production follows predicted metalation of $Mg^{II}GTP$-CobW. a** Predicted metalation of $Mg^{II}GTP$-CobW with $Co^{II}$ and $Zn^{II}$ (open and grey bars, respectively) in samples treated with defined media $[Co^{II}]$. Intracellular $\Delta G_{Co(II)}$ for each condition was calculated from *rcnA* expression (Fig. 8 and Supplementary Table 8). **b** $B_{12}$ produced by *E. coli** strains with and without *cobW* (open and grey bars, respectively) following 4 h exposure to defined $[Co^{II}]$. $B_{12}$ was detected using a *Salmonella* AR2680 bioassay[65] (detects corrins, expected to be predominantly $B_{12}$; see "Methods") and quantified by automated analysis of growth areas (Supplementary Fig. 29 and Supplementary Software 1). Inset shows original image and detected areas (each false coloured) for representative ($n = 3$) bioassay plate of $B_{12}$ calibration standards. All data are the mean ± SD of $n = 3$ biologically independent replicates (with errors in **a** propagated from qPCR data in Fig. 8b). Triangles represent individual experiments (some data points overlap, experimental values are available in Source Data files).

mechanism. $Zn^{II}$ is identified as the preferred metal for nucleotide-bound forms of YeiR and YjiA: this is due to their weaker affinities for $Co^{II}$ relative to $Mg^{II}GTP$-CobW (Fig. 7 and Supplementary Table 5), rather than tighter affinities for $Zn^{II}$. These data illustrate the value of using the metalation calculator provided as Supplementary Data 1, which can now be broadly applied to metal-speciation in the context of intracellular competition.

$Mg^{II}GTP$-CobW binds $Zn^{II}$ and $Cu^{I}$ more tightly than $Co^{II}$ (Figs. 2e, 3, 4 and Supplementary Table 2), and likewise nucleotide-bound forms of YeiR and YjiA bind $Cu^{I}$ more tightly than $Zn^{II}$. Notably, by taking into account intracellular metal availability, $\Delta\Delta G$ for $Cu^{I}$ was shown to be greater than zero for all three proteins in idealised cells, and also in conditional cells at either 90% or 99% of the dynamic range of the $Cu^{I}$ sensor CueR (Figs. 5, 6 and Supplementary Fig. 14). Thus, these proteins will not acquire $Cu^{I}$. However, for $Mg^{II}GTP$-CobW $\Delta\Delta G$ for $Zn^{II}$ was below zero in an idealised cell suggesting mis-metalation with

$Zn^{II}$ (Fig. 5). Indeed, given that CobW binds $Zn^{II}$ more tightly than many known $Zn^{II}$ proteins[14,43], and comparably to YeiR and YjiA (Supplementary Tables 2 and 5), it seems remarkable that $Zn^{II}$ is not the cognate metal. The data in Fig. 4, plus Supplementary Table 4, illustrate how occupancies of $Mg^{II}GTP$-CobW with $Co^{II}$ versus $Zn^{II}$ change as a function of relative buffered metal availabilities. By reference to intracellular available free energies, the metal with the most negative $\Delta\Delta G$ will have the highest occupancy in vivo (Eq. (4)). In an idealised cell, $\Delta\Delta G$ for $Co^{II}$ is more negative than $\Delta\Delta G$ for $Zn^{II}$ and so the weaker binding metal dominates (Fig. 5 and Supplementary Table 2). In contrast, for nucleotide-bound forms of YeiR and YjiA $\Delta\Delta G$ for $Zn^{II}$ is more negative than $\Delta\Delta G$ for $Co^{II}$ making these deduced $Zn^{II}$ proteins. The previously intractable challenge to understand inter-metal competition in a cell now becomes tractable.

Initial calculations here, and in previous work[9], assume an idealised cell in which the metal sensors are at the mid-points of their dynamic ranges ($\theta_D = 0.5$). Therefore, we have calculated the available $\Delta G_{Co(II)}$ in real (conditional) cells from the responses of RcnR ($\theta_D$) estimated experimentally by qPCR of $rcnA$ (Fig. 8 and Supplementary Fig. 27). As with other metallochaperones[10,44], CobW is crucial when the cognate metal is limiting but at elevated $Co^{II}$, CobW-independent synthesis of $B_{12}$ occurs (Fig. 9b). CobNST must acquire $Co^{II}$ directly from the cytosol at the higher available $\Delta G_{Co(II)}$. Importantly, CobW-dependent $B_{12}$ synthesis tracked with the calculated $Co^{II}$ occupancy of $Mg^{II}GTP$-CobW in cells supplemented with different amounts of $Co^{II}$ (Fig. 9). By monitoring the responses of sensors for different metals, it will be possible to define available $\Delta G$, and predict protein occupancies with diverse metals, in different growth conditions. The calculator should be most accurate in *Salmonella* and closely related species such as *E. coli*. However, metal availabilities can also be adjusted (and/or simulated) to account for species differences, noting that the dynamic ranges of available $\Delta G$ values might be similar even when total cellular metal changes greatly between species.

Spectral features indicate that the $Co^{II}$ site in $Mg^{II}GTP$-CobW involves thiols, likely derived from the CxCC motif in the GTPase domain, and a tetrahedral geometry (Figs. 1, 2 and Supplementary Fig. 3). However, all COG0523 proteins contain the CxCC motif[16] and there is now a quest to understand why $Co^{II}$ affinities are weaker for $Mg^{II}GTP$-bound YeiR and YjiA (and, hypothetically, ZigA and ZagA), which bind predominantly to $Zn^{II}$ as a result (Fig. 6). Notably a further pair of conserved cysteine residues ($C_{56}$, $C_{61}$) in CobWs are absent from the homologues (Supplementary Fig. 31). Intriguingly, $Ni^{II}$ binding to $Mg^{II}GTP$-CobW, $Mg^{II}GTP$-YeiR and $Mg^{II}GTP\gamma S$-YjiA does not follow the order of stabilities of metal binding predicted by the Irving-Williams series (Figs. 5 and 6). An appealing explanation is that the allosteric coupling of GTP- and metal-binding imposes a (tetrahedral) geometry on the metal site that would disfavour $Ni^{II}$ coordination (the Irving-Williams series applies where there is no steric selection): notably, related G3E GTPases involved in $Ni^{II}$ homoeostasis (HypB and UreG) display square planar $Ni^{II}$ coordination geometry[45,46].

The metalation calculator has identified cognate metals for three members of the COG0523 sub-family of GE3 GTPases (Figs. 5, 6 and Table 1), and this work establishes vitamin $B_{12}$ as the ultimate $Co^{II}$-client of CobW (Fig. 9). For YeiR and YjiA, there is now a quest to identify their distinct roles and potential $Zn^{II}$-requiring client(s). Under-metalation of $Mg^{II}GTP$-CobW with $Co^{II}$ (and resultant mis-metalation with $Zn^{II}$, Fig. 9a) could be especially problematic in *E. coli*\* due to the lack of a dedicated $Co^{II}$ import system in this bacterium[47]. This suggests tantalising opportunities to engineer strains suited to the manufacture of vitamin $B_{12}$, either via enhanced $Co^{II}$ uptake through engineered

$Co^{II}$-import, or via impaired $Zn^{II}$ accumulation by endogenous $Zn^{II}$-transport systems. By analogy, with almost a half of enzymes requiring metals, an ability to calculate metalation in vivo should have broad applicability in optimising (or subverting) metalation in biotechnology. The calculator (Supplementary Data 1) can be widely used to understand metalation and mismetalation of proteins that acquire $Mg^{II}$, $Mn^{II}$, $Fe^{II}$, $Co^{II}$, $Ni^{II}$, $Cu^{I}$ or $Zn^{II}$ from the milieu inside living cells.

## Methods

**Protein expression and purification**. The DNA sequence coding CobW was amplified by PCR using primers 1 and 2 (Supplementary Table 10) with genomic DNA from *R. capsulatus* SB1003 as template. The amplified fragment contained an NdeI restriction site at the 5′ end and a SpeI site at the 3′ end, allowing it to be cloned into a modified pET-3a vector[34]. The DNA coding sequence of *yeiR* (SL1344_2189) from *S. enterica* serovar Typhimurium strain SL1344 was obtained as a synthetic gene from Eurofins in a pEX plasmid with the T7 promoter and terminator from pET29a flanking *yeiR*. Additionally, the start codon of the gene (GTG) was changed to the more common ATG (pEX*yeiR*). The native coding sequence of *yjiA* (SL1344_4461) was obtained in the same manner (pEX*yjiA*). The correct sequence of each gene (*cobW*, *yeiR* and *yjiA*) was confirmed by DNA sequencing (DBS Genomics – Durham University).

*E. coli* BL21(DE3) pLysS, transformed with either pET3a-*cobW*, pEX-*yeiR* or pEX-*yjiA* were cultured in LB medium with antibiotics carbenicillin (50–100 mg L$^{-1}$) and chloramphenicol (30–34 mg L$^{-1}$). At mid-log phase, protein expression was induced by addition of 0.4 mM (CobW), 0.5 mM (YjiA) or 1.0 mM (YeiR) IPTG. Cells were cultured (with shaking) for 3–4 h at 37 °C (CobW) or at 20 °C overnight (YeiR and YjiA). Cells were harvested and stored at −20 °C prior to use.

Cells overexpressing CobW were resuspended in 20 mM sodium phosphate pH 7.4, 500 mM NaCl, 5 mM imidazole, 5 mM DTT and 1 mM PMSF for lysis (sonication) and cell debris was pelleted by centrifugation (38,000 × *g*, 45 min, 4 °C). Lysate was loaded to a 5-mL HisTrap HP column (GE Heathcare) pre-equilibrated in suspension buffer. CobW binds to the HisTrap column courtesy of a natural His-rich region within the protein. The column was washed with suspension buffer (10 column volumes), then eluted with suspension buffer containing 100 mM imidazole. Protein-containing fractions were incubated with excess (≥10-fold) EDTA for ≥1 h before being loaded to a HiLoad 26/600 Superdex 75 size exclusion column equilibrated in 50 mM Tris pH 8.0, 150 mM NaCl, 5 mM DTT and eluted in the same buffer. Peak CobW-containing fractions (determined from SDS-PAGE) were pooled, concentrated to ~0.5 mL (using a Vivaspin® 15 Turbo centrifugal concentrator). Protein identity was confirmed using ESI-MS by Durham University Department of Chemistry Mass Spectrometry Service. ESI-MS data were recorded on a QtoF Premier mass spectrometer coupled to an Acuity UPLC system (Waters). Protein samples were desalted prior to injection using a Waters MassPrep desalting cartridge (2.1 × 10 mm) and eluted with a linear acetonitrile gradient (20–80% v/v; 0.1% formic acid). Spectra were processed using Masslynx 4.1, deconvoluted using MaxEnt 1 and data imported into SigmaPlot software for preparation of figures.

Cells overexpressing YeiR were resuspended in 20 mM sodium phosphate pH 7.4, 100 mM NaCl, 5 mM DTT, 1 mM PMSF for lysis (sonication) and cell debris was pelleted by centrifugation (31,191 × *g*, 15 min, 8 °C). Soluble lysate was applied to a 5-mL HisTrap column (GE Healthcare) equilibrated with lysis buffer without PMSF. The column was washed with equilibration buffer before elution with equilibration buffer containing 10, 50 and 100 mM imidazole. YeiR eluted in the buffer containing 50 mM imidazole. EDTA was added to the YeiR-containing fraction to a final concentration of 10 mM and stored overnight at 4 °C. The sample was applied to HiLoad 26/600 Superdex 75 (GE Healthcare) equilibrated with 50 mM Tris pH 8, 150 mM NaCl, 5 mM DTT and eluted with the same buffer. Peak fractions were pooled and applied to a 5-mL Q anion exchange column (GE Healthcare) equilibrated with the size exclusion column buffer. Column flow through and wash were collected before eluting the column with size exclusion column buffer with the addition of 1 M NaCl. YeiR displays no affinity for the Q column and elutes in the flow through and wash. The remaining major contaminant elutes with 1 M NaCl. The flow through and wash from the Q column were pooled and concentrated using a Spin-X UF concentrator (Corning, 10 kDa molecular weight cut-off).

Cells overexpressing YjiA were resuspended in 20 mM Tris 7.5, 100 mM NaCl, 5 mM DTT, 1 mM PMSF for lysis (sonication) and cell debris was pelleted by centrifugation (two consecutive 20 min runs, 39,191 × *g*, 4 °C) before passing clarified supernatant through a 20-μm nylon membrane filter. Soluble lysate was applied to a 5-mL HisTrap column (GE Healthcare) equilibrated with lysis buffer without PMSF, collecting the flow through and a one column volume wash. This step removes a major contaminant which binds to the HisTrap column. Pooled flow through and wash fractions were applied to a 5-mL HiTrap Q-Sepharose fast flow column (GE Healthcare) equilibrated with lysis buffer without PMSF. The column was washed with equilibration buffer then equilibration buffer with 200 mM NaCl before elution of YjiA by application of a 50-mL gradient of 200–600 mM NaCl in equilibration buffer collecting 5 mL fractions. YjiA typically eluted

between 10 and 30 mL. Fractions containing the highest concentration of YjiA with the lowest degree of contamination, as judged by SDS-PAGE, were stored overnight (4 °C) with EDTA to a final concentration of 5 mM. Fractions were concentrated to 5 mL using a Spin-X UF concentrator (Corning, 10 kDa molecular weight cut-off) and applied to HiLoad 26/600 Superdex 75 (GE Healthcare) equilibrated with lysis buffer without PMSF and eluted with the same buffer. Fractions were pooled and concentrated to 5–20 mg mL$^{-1}$ using a Spin-X UF concentrator (Corning, 10 kDa molecular weight cut-off) before storage at −80 °C.

Following purification, CobW, YeiR and YjiA samples were transferred to an anaerobic glovebox (Belle Technology), (0.5–1 mL) applied to a PD-10 Desalting Column prepacked with Sephadex G-25 medium (GE Healthcare) equilibrated with chelex-treated and N$_2$-purged buffer (10 mM HEPES pH 7.0, 100 mM NaCl, 400 mM KCl) and eluted in the same buffer. Proteins were quantified by $A_{280 nm}$ using experimentally determined extinction coefficients ($ε = 15,300$ M$^{-1}$ cm$^{-1}$ for CobW, 52,745 M$^{-1}$ cm$^{-1}$ for YeiR, and 27,900 M$^{-1}$ cm$^{-1}$ for YjiA) determined by quantitative amino acid analysis (Alta Bioscience Ltd). Samples were confirmed to be of high purity by SDS-PAGE (full gel images are available in the Supplementary Information), and ≥92.5% (YeiR) and ≥95% (CobW, YjiA) metal-free (by inductively coupled plasma-mass spectrometry; ICP-MS). ICP-MS of one sample of YjiA was performed 3 weeks after buffer exchange and >5% zinc detected, but results were consistent with replicates containing <5% zinc. ICP-MS was conducted using Durham University Bio-ICP-MS Facility (PlasmaLab software; Thermo Fisher). Reduced thiol content was determined by reaction with ~10-fold excess of Ellman's reagent 5,5′-dithio-bis-[2-nitrobenzoic acid] (DTNB; produces one equivalent of chromophore TNB$^{2-}$ per protein thiol, $A_{412 nm} = 14,150$ M$^{-1}$ cm$^{-1}$)[48,49]. For CobW >5.5 cysteines were found to be reactive with DTNB (expected value = 6), for YeiR 4–5 cysteines (expected value = 5), and for YjiA >4.5 cysteines (expected value = 5).

**Preparation of metal stocks**. All metal stocks were prepared in ultrapure water from appropriate salts (MgCl$_2$, (NH$_4$)$_2$Fe(SO$_4$)$_2$, CoCl$_2$, NiSO$_4$, NiCl$_2$, CuSO$_4$, ZnCl$_2$, ZnSO$_4$) and quantified by ICP-MS analysis. Fe$^{II}$ stocks were prepared by dissolving (NH$_4$)$_2$Fe(SO$_4$)$_2$·6H$_2$O in deoxygenated 0.1% (v/v) HCl in an anaerobic chamber. Reaction with excess ferrozine (Fz; ~50-fold) confirmed that iron was ≥95% reduced (Fe$^{II}$Fz$_3$ $ε_{562 nm} = 27,900$ M$^{-1}$ cm$^{-1}$)[50]. Concentrated stocks were diluted daily in deoxygenated ultrapure water to prepare working solutions and confirmed to be ≥90% Fe$^{II}$. Other metal stocks were prepared aerobically as concentrated stocks and diluted to working solutions with deoxygenated ultrapure water in an anaerobic chamber. Cu(I) was generated in situ (from CuSO$_4$) by hydroxylamine (1–10 mM) which quantitatively reduces Cu(II) to Cu(I) in the presence of excess chelator (L = Bca or Bcs) to form Cu$^I$L$_2$ complexes[51].

**Determination of CobW Co$^{II}$-binding stoichiometries**. Metal-binding experiments were conducted in an anaerobic chamber in deoxygenated, chelex-treated 10 mM HEPES pH 7.0, 100 mM NaCl, 400 mM KCl. For stoichiometry determinations, Co$^{II}$ was titrated into a solution of CobW (15–30 μM) together with relevant nucleotides (supplied in ~10-fold excess of protein concentration for GTP and GDP and ~3-fold excess for GMPPNP and GTPγS, as specified in figure legends) in the absence or presence of Mg$^{II}$ (2.7 mM). Absorbance was recorded using a Lambda 35 UV-visible spectrophotometer (Perkin Elmer; UV-Win lab software). The extinction coefficient of Co$^{II}$Mg$^{II}$GTP-CobW ($ε_{339 nm} = 2800 ± 100$ M$^{-1}$ cm$^{-1}$, average ± s.d of $n = 3$ independent titrations) was determined from absorbance at saturating metal concentrations (Supplementary Fig. 3g). Extinction coefficients of related complexes Co$^{II}$Mg$^{II}$GMPPNP-CobW, Co$^{II}$Mg$^{II}$GTPγS-CobW, Co$^{II}$GTP-CobW, Co$^{II}_2$GMPPNP-CobW and Co$^{II}_2$GTPγS-CobW were similarly determined (Figs. 1, 2 and Supplementary Figs. 1, 3): within experimental error, all produced the same extinction coefficient as for Co$^{II}$Mg$^{II}$GTP-CobW, thus $ε_{339 nm} = 2800$ M$^{-1}$ cm$^{-1}$ was assigned to all species. Gel-filtration chromatography experiments were performed by incubating CobW (10 μM) and Co$^{II}$ (30 μM) for 30 min with or without cofactor GMPPNP (30 μM) then applying 0.5 mL to a PD-10 Sephadex G-25 gel-filtration column (GE Healthcare). Eluted fractions (0.5 mL) were analysed for cobalt by ICP-MS and for protein by Bradford assay.

**Determination of CobW metal affinities via ligand competition**. Ligand competition experiments were conducted in an anaerobic chamber in deoxygenated, chelex-treated 10 mM HEPES pH 7.0, 100 mM NaCl, 400 mM KCl, except where high concentrations (≥1 mM) of competing ligand were employed, where 50 mM HEPES was used to maintain buffered pH 7.0. Absorbance was recorded using a Lambda 35 UV-visible spectrophotometer (Perkin Elmer). Fluorescence spectra were recorded using a Cary Eclipse fluorescence spectrophotometer (Agilent; Cary Eclipse scan application software). Affinities were determined at a range of different competing ligand and/or the protein:ligand ratio) to ensure reliability: details are documented in Supplementary Table 1. Scripts used for data fitting (using Dynafit[52]) are provided in Supplementary Software 2. The effect of Mg$^{II}$ (2.7 mM) on apparent dissociation constants of ligand standards (EGTA, NTA, fura-2, Mf2 and quin-2) was calculated to be insignificant under the conditions of competition experiments (Supplementary Table 3). For probes with undefined Mg$^{II}$ affinities (Tar, Bca), control experiments

confirmed that addition of Mg$^{II}$ (2.7 mM) had negligible effect on competition experiments (Supplementary Figs. 10d and 12). Thus, Mg$^{II}$ was not incorporated into the curve-fitting models.

For determination of weaker ($K_D > 10$ nM) Co$^{II}$ binding affinities (CobW and Mg$^{II}$GDP-CobW), CoCl$_2$ was titrated into a solution of 5-Oxazolecarboxylic acid, 2-(6-(bis(carboxymethyl)amino)-5-(2-(6-(bis(carboxymethyl)amino)-5-methylphenoxy)ethoxy)-2-benzofuranyl)-pentapotassium salt (fura-2; quantified by $ε_{363 nm} = 28,000$ M$^{-1}$ cm$^{-1}$)[53] and CobW in the presence or absence of cofactors (MgCl$_2$ and GDP) and fluorescence emission ($λ_{ex} = 360$ nm; $λ_{max}$ ~505 nm) was recorded at equilibrium. Co$^{II}$-dependent fluorescence quenching of fura-2 was used to determine Co$^{II}$ speciation. For determination of Co$^{II}$ binding affinities tighter than 10 nM (Mg$^{II}$GMPPNP-CobW, Mg$^{II}$GTPγS-CobW and Mg$^{II}$GTP-CobW), CoCl$_2$ was titrated into a solution containing CobW, competing ligand (EGTA or NTA), MgCl$_2$ and nucleotide (GMPPNP, GTPγS or GTP). UV-visible absorbance (relative to metal-free solution) was recorded at equilibrium to determine Co$^{II}$ speciation ($ε_{339 nm} = 2800$ M$^{-1}$ cm$^{-1}$ for Co$^{II}$-bound proteins). Data were fit using Dynafit[52] to models describing 1:1 binding stoichiometry for Co$^{II}$:protein and 1:1 binding stoichiometry for Co$^{II}$:ligand (ligand = fura-2, EGTA or NTA). Ligand dissociation constants at pH 7.0: fura-2 $K_{Co(II)} = 8.6 × 10^{-9}$ M (ref. [39]); EGTA $K_{Co(II)} = 7.9 × 10^{-9}$ M (ref. [38]); NTA $K_{Co(II)} = 2.2 × 10^{-8}$ M (ref. [38]).

(NH$_4$)Fe(SO$_4$)$_2$ was titrated into a solution of Tar (16 μM), MgCl$_2$ (2.7 mM) and GTP (500 μM) in the absence or presence of CobW (50 μM) and UV-visible absorbance recorded at equilibrium to define Fe$^{II}$ speciation (Fe$^{II}$Tar$_2$ $ε_{720} = 19,560$ M$^{-1}$ cm$^{-1}$ under experimental conditions, Supplementary Fig. 8a). Data were fit in Dynafit[52] to a model describing 1:1 binding stoichiometry for Fe$^{II}$:protein and 1:2 binding stoichiometry for Fe$^{II}$:Tar using $β_{2,Fe(II)} = 4.0 × 10^{13}$ M$^{-2}$ for Tar at pH 7.0 (ref. [54]). Experimental data were compared to simulated fits with defined protein $K_{Fe(II)} = 10^{-6}$ M, $10^{-7}$ M, allowing limiting $K_D ≥ 10^{-6}$ M for Mg$^{II}$GTP-CobW to be determined. Tar stock concentrations were quantified using $ε_{470 nm} = 24,800$ M$^{-1}$ cm$^{-1}$ (reported value at pH 7.0 (ref. [54])) and verified by titration with metal stocks (Fe$^{II}$ or Ni$^{II}$, quantified by ICP-MS).

NiSO$_4$ was titrated into a solution of Tar (20 μM), CobW (10–30 μM), MgCl$_2$ (2.7 mM) and GTP (100–300 μM) and UV-visible absorbance recorded at equilibrium to determine Ni$^{II}$ speciation (Ni$^{II}$Tar$_2$ $Δε_{535 nm} = 3.8 (±0.1) × 10^4$ M$^{-1}$ cm$^{-1}$ relative to ligand only solution; Supplementary Fig. 10a). Tar stock concentrations were quantified as above. Data were fit using Dynafit[52] to a model describing 1:1 stoichiometry Ni$^{II}$:protein and 1:2 stoichiometry Ni$^{II}$:Tar; $β_{2,Ni(II)} = 4.3 (±0.6) × 10^{15}$ M$^{-2}$ for Tar at pH 7.0 was independently determined by preparing a series of solutions of NiTar$_2$ ([Ni$^{II}$] = 15 μM, [Tar] = 36 μM) with varying EGTA concentrations (0–400 μM) and measuring UV-visible absorbance at equilibrium (following 1–2 h incubation). EGTA $K_{Ni(II)} = 5.0 × 10^{-10}$ M at pH 7.0 (ref. [38]). Data were fit to Eq. (5)[51] using Kaleidagraph (Synergy Software).

$$\frac{[EGTA]_{tot}}{[Ni^{II}]_{tot}} = 1 - \frac{[Ni^{II}Tar_2]}{[Ni^{II}]_{tot}} + K_D(EGTA)β_2(Tar)\left(\frac{[Tar]_{tot}}{[Ni^{II}Tar_2]} - 2\right)^2 [Ni^{II}Tar_2]\left(1 - \frac{[Ni^{II}Tar_2]}{[Ni^{II}]_{tot}}\right)$$

(5)

CuSO$_4$ was titrated into a solution of Bca (1.0 mM), CobW (10–30 μM), MgCl$_2$ (2.7 mM), GTP (100–300 μM) and reductant NH$_2$OH (1.0 mM) which quantitatively reduces Cu$^{II}$ to Cu$^I$ in the presence of a strong Cu$^I$ ligand (e.g. Bca: $β_{2,Cu(I)} = 1.6 × 10^{17}$ M$^{-2}$ (ref. [38])). UV-visible absorbance was recorded at equilibrium to define Cu$^I$ speciation (Cu$^I$Bca$_2$ $ε_{562} = 7900$ M$^{-1}$ cm$^{-1}$ (ref. [38])) and data were fit using Dynafit[52] to a model describing 1:1 stoichiometry Cu$^I$:protein and 1:2 stoichiometry Cu$^I$:Bca.

ZnCl$_2$ was titrated into a solution containing quin-2 (10 μM), CobW (10 μM), MgCl$_2$ (2.7 mM) and GTP (50 μM) and UV-visible absorbance recorded at equilibrium. Quin-2 was quantified using $ε_{261 nm} = 37,000$ M$^{-1}$ cm$^{-1}$ (ref. [55]). $K_{Zn(II)}$ for Mg$^{II}$GTP-CobW was beyond the range of this experiment (significantly tighter than quin-2) and only a limiting affinity was determined ($K_{Zn(II)} < 10^{-12}$ M).

**Zn$^{II}$ affinity of Mg$^{II}$GTP-CobW via inter-metal competition**. Solutions containing CobW (17.9–20.4 μM), MgCl$_2$ (2.7 mM), GTP (200 μM) and ligand NTA (0.4–4.0 mM) were titrated with CoCl$_2$ (0.3–3.0 mM) and ZnCl$_2$ (15.3–25.5 μM) and UV-visible absorbance was recorded at equilibrium to determine Co$^{II}$ occupancy of CobW ($ε_{339 nm} = 2800$ M$^{-1}$ cm$^{-1}$ for Co$^{II}$Mg$^{II}$GTP-CobW). Details of individual experiments are in Supplementary Table 4. The total concentration of Co$^{II}$ and Zn$^{II}$ in each solution was limiting, such that both metals were buffered by ligand NTA. Metal speciation was determined from the mass balance relationships given in Eqs. (6–8) (cofactors Mg$^{II}$GTP omitted for clarity). Thus, $K_{Zn(II)}$ for Mg$^{II}$GTP-CobW was calculated from the exchange equilibria ($K_{ex}$) in Fig. 4a, relative to known $K_{Co(II)}$ for the protein (Supplementary Table 2) and ligand dissociation constants (NTA $K_{Zn(II)} = 1.18 × 10^{-8}$ M, $K_{Co(II)} = 2.24 × 10^{-8}$ M (ref. [38])). These calculations are valid given that [M]$_{free}$ ≪ [M]$_{tot}$ (M = Co$^{II}$ or Zn$^{II}$, buffered by excess NTA), the concentration of non-metalated protein is negligible (Supplementary Fig. 13) and potential ternary complexes involving metal, protein and NTA are transient species only with insignificant concentration at thermodynamic equilibrium (varying ratios of buffered metals, [Co$^{II}$NTA]/[Zn$^{II}$NTA], were used to confirm consistency of $K_D$ values at multiple equilibria; see Fig. 4 and

Supplementary Table 4).

$$[Co^{II}NTA] = [Co^{II}]_{tot} - [Co^{II}CobW] \qquad (6)$$

$$[Zn^{II}CobW] = [CobW]_{tot} - [Co^{II}CobW] \qquad (7)$$

$$[Zn^{II}NTA] = [Zn^{II}]_{tot} - [Zn^{II}CobW] \qquad (8)$$

**Determination of YeiR and YjiA metal stoichiometries and affinities.** Investigation of protein–metal interactions and competition experiments to determine metal affinities were performed in 10 mM HEPES pH 7, 100 mM NaCl, 400 mM KCl (chelex treated and $N_2$ purged) with the inclusion of nucleotides and $MgCl_2$ as noted in figure legends. Absorbance was recorded using a Lambda 35 UV-visible spectrophotometer (Perkin Elmer). Fluorescence spectra were recorded using a Cary Eclipse fluorescence spectrophotometer (Agilent). Scripts used for data fitting (using Dynafit[52]) are provided in Supplementary Software 2.

$CoCl_2$ was titrated into a solution of fura-2 ($\varepsilon_{363\,nm} = 28,000$ $M^{-1}$ $cm^{-1}$, $K_{Co(II)}$ $= 8.6 \times 10^{-9}$ M (refs. [39,53])) in the presence of YeiR or YjiA and fluorescence emission (510 nm) recorded at equilibrium ($\lambda_{ex} = 360$ nm, 20 °C). Data were fit to a model describing 1:1 $Co^{II}$:fura-2 and 1:1 $Co^{II}$:protein binding stoichiometries using Dynafit[52].

$NiCl_2$ was titrated into a solution of Mf2 ($\varepsilon_{369\,nm} = 22,000$ $M^{-1}$ $cm^{-1}$, $K_{Ni(II)} = 5 \times 10^{-8}$ M (refs. [53,56])) in the presence of YeiR or YjiA and the absorbance (323–325 and 365–366 nm) recorded at equilibrium. Data were fit (both wavelengths simultaneously) to a model describing 1:1 $Ni^{II}$:Mf2 and 1:1 $Ni^{II}$: protein binding stoichiometries using Dynafit[52].

$ZnSO_4$ was titrated into a solution of Mf2 ($K_{Zn(II)} = 2 \times 10^{-8}$ M (ref. [57])), PAR ($\beta_{2\,Zn(II)} = 2 \times 10^{12}$ $M^{-2}$ (ref. [58])) or quin-2 ($K_{Zn(II)} = 3.7 \times 10^{-12}$ M (ref. [55])) in the presence of YeiR or YjiA and the absorbance (325 and 366 nm Mf2; 500 nm PAR; 265 or 269 nm quin-2) recorded at equilibrium. Concentrations of PAR and quin-2 stocks were determined by direct titration with $ZnSO_4$. Data were fit to a model describing 1:1 $Zn^{II}$:quin-2 and 1:1 $Zn^{II}$:YjiA binding stoichiometries. Zn:YeiR stoichiometries were fit as 1:1, or allowed to be determined in fitting as described in the text using Dynafit[52].

$CuSO_4$ was titrated into a solution of Bca ($Cu^{I}Bca_2$ $\varepsilon_{562\,nm} = 7900$ $M^{-1}$ $cm^{-1}$, $\beta_2$ $Cu(I) = 10^{17.2}$ $M^{-2}$ (ref. [59])) in the presence and absence of YeiR or YjiA (with inclusion of hydroxylamine) and absorbance (562 nm) recorded at equilibrium. Protein $Cu^{I}$ affinity was calculated using Eq. (9), for the tightest binding event. Calculated affinities were simulated using Dynafit[52], and overlaid on the data.

$$K_D\beta_2 = \frac{\left(\frac{[P]_{tot}}{[MP]}\right) - 1}{\left(\left(\frac{[L_{tot}]}{[ML_2]} - 2\right)^2 [ML_2]\right)} \qquad (9)$$

$(NH_4)Fe(SO_4)_2$ was titrated into a solution of Tar ($Tar_2Fe(II)$ $\varepsilon_{720\,nm} = 19,000$ $M^{-1}$ $cm^{-1}$, $\beta_2$ $Fe(II) = 10^{13.6}$ $M^{-2}$ (at pH 7.0) (ref. [54])) in the presence and absence of YeiR or YjiA and absorbance (720 nm) recorded at equilibrium. Data were fit to a model describing 1:2 $Fe^{II}$:Tar and 1:1 $Fe^{II}$:protein binding stoichiometries using Dynafit[52].

$MnCl_2$ was titrated into a solution of Mf2 ($K_{Mn(II)} = 6.1 \times 10^{-6}$ M (ref. [9])) in the presence of YjiA and the absorbance (330 and 365 nm) recorded at equilibrium. Data were fit (both wavelengths simultaneously) to a model describing 1:1 $Mn^{II}$: Mf2 and 1:1 $Mn^{II}$:protein binding stoichiometries using Dynafit[52].

Gel filtration chromatography of YeiR was performed by application of 0.5 mL (10 μM) to a PD-10 Desalting Column prepacked with Sephadex G-25 medium equilibrated with buffer supplemented with 2.7 mM $MgCl_2$ with or without 20 μM $MnCl_2$ and eluted with the same buffer. YeiR was incubated with 20 μM $MnCl_2$ for 20 min prior to application to the column. Protein content of collected fractions was assayed by $A_{280\,nm}$ and Bradford assay, metal content by ICP-MS.

**Inter-protein competition for Co(II).** Experiments were performed in an anaerobic glovebox. YeiR (10 μM) was incubated with GTP (100 μM), $MgCl_2$ (2.7 mM) and $CoCl_2$ (8 μM) in 10 mM HEPES pH 7.0, 40 mM NaCl, 160 mM KCl (chelex treated and $N_2$ purged) for 10 min before addition of CobW (10 μM) (total volume upon CobW addition = 1.1 mL). The mixture was incubated for a further 30 min before application of 1 mL of the incubation reaction to a 1-mL Q anion exchange column (GE Healthcare) equilibrated with 10 mM HEPES pH 7.0, 40 mM NaCl, 160 mM KCl (chelex treated and $N_2$ purged), collecting the flow through. The column was sequentially eluted with equilibration buffer collecting six 0.5 mL fractions followed by 10 mM HEPES pH 7.0, 200 mM NaCl, 800 mM KCl (chelex treated and $N_2$ purged) collecting six 0.5-mL fractions. Fractions were analysed for protein content by SDS-PAGE and for metal content by ICP-MS. Controls were conducted concurrently as above but with YeiR or CobW alone.

**GTPase activity assays.** CobW (20–50 μM) was incubated with $CoCl_2$ (0.9 equivalents $Co^{II}$:protein) and GTP (200 μM) in an anaerobic chamber in $N_2$-purged, chelex-treated 10 mM HEPES pH 7.0, 100 mM NaCl, 400 mM KCl. Aliquots of solution taken at various time intervals (0–390 min) were loaded to a 5-mL HiTrap Q HP column (GE Healthcare) equilibrated in buffer (20 mM HEPES pH 7.0, 100 mM NaCl) and eluted with a linear NaCl gradient (100–500 mM NaCl).

Nucleotides were detected by UV absorbance (254 nm or 280 nm) and the ratio of GTP:GDP in each sample was calculated by integration of the respective peak areas.

**Growth of E. coli\* strains.** E. coli\* strains used in this work are derived from E. coli MG1655 (DE3) engineered to contain a set of $B_{12}$ biosynthesis genes from R. capsulatus[60,61], and Brucella melitensis (B. melitensis)[34]. Strain ED741 (E. coli\* without cobW) is MG1655 with $P_{lac}$-T7RNAP-$P_{T7}$-cobAIGJFMKLHBROQJD-bluE-C-bluF-PUB-cbiW-VE-$P_{T7}$-cobNST while strain ED732 (E. coli\* with cobW) is MG1655 with $P_{lac-T7}$RNAP-$P_{T7}$-cobAIGJFMKLHBROQJD-bluE-C-bluF-PUB-cbiW-VE-$P_{T7}$-cobWNST. All R. capsulatus and B. melitensis (cobG, cobR, cobE) genes were cloned individually in pET3a and subcloned together using the link and lock method[34]. The synthetic operons were transferred into the E. coli genome using CRISPR technology[62]. Although chromosomally integrated $B_{12}$ biosynthesis genes are IPTG-inducible under the control of the T7 promoter, in the current experiments IPTG was not added to cell cultures to avoid potential disruptions of cellular metal homoeostasis caused by over-production of metalloproteins.

All cultures and media were prepared in plasticware or acid-washed glassware to minimise trace metal contamination. LB medium was inoculated with overnight culture of E. coli\* ($OD_{600\,nm} = 0.025$) and incubated at 37 °C with shaking until $OD_{600\,nm}$ reached ~0.2. Aliquots (5 mL or 50 mL) of this culture were treated with sterile $CoCl_2$, $H_2O$, EDTA or $ZnCl_2$ (100× concentrated stocks) to reach final concentrations as specified in figure legends (Figs. 8b, 9b and Supplementary Figs. 27, 28a, b, d and 29c) and incubated under the same conditions for a further 1–4 h. Samples used for RNA extraction were taken 1 h after treatment. Samples for $B_{12}$ quantification and $OD_{600\,nm}$ readings were taken 4 h after treatment to ensure detectable corrinoid production.

**Determination of transcript abundance in E. coli\*.** Aliquots (1 mL) of E. coli\* culture from each growth condition were stabilised in RNAProtect Bacteria Reagent (2 mL; Qiagen) and cells pellets were frozen at −80 °C prior to processing. RNA was extracted using an RNeasy Mini Kit (Qiagen) as described by the manufacturer. RNA was quantified by absorbance at 260 nm and treated with DNAse I (2.5 U/μL; Fermentas). cDNA was generated using the ImProm-II Reverse Transcriptase System (Promega) with 300 ng RNA per reaction, and control reactions without reverse transcriptase were conducted in parallel. Transcript abundance was determined using primers 3 and 4 for rcnA, 5 and 6 for zntA, 7 and 8 for znuA, 9 and 10 for rpoD, each pair designed to amplify ~110 bp fragment. Quantitative PCR analysis was carried out in 20 μL reactions using 5 ng of cDNA, 0.8 μM of each appropriate primer and PowerUp SYBR Green Master Mix (Thermo Fisher Scientific). Three technical replicates of each sample (i.e. biological replicate) were analysed using a Rotor-Gene Q 2plex (Qiagen; Rotor-Gene-Q Pure Detection software), plus control reactions without cDNA template for each primer pair. The fold change, relative to the mean of the control condition for each sensor, was calculated using the $2^{-\Delta\Delta CT}$ method[63], with rpoD as the reference gene. $C_q$ values were calculated with LinRegPCR after correcting for amplicon efficiency[64].

**Intracellular available $\Delta G_{Co(II)}$ under bespoke conditions.** Intracellular available $\Delta G_{metals}$ were first calculated from available metal concentrations where the cognate sensor is at 1%, 10%, 50%, 90% and 99% of its response (i.e., $\theta_D = 0.01, 0.1, 0.5, 0.9, 0.99$; Supplementary Note 1). Available metal concentrations corresponding to these fractional occupancies were determined using known metal affinities, DNA affinities, protein abundances and numbers of DNA binding sites determined for Salmonella sensors[9], using excel spreadsheet (Supplementary Dataset 1) and MATLAB code (Supplementary Note 3) available in ref. [9].

Fractional responses ($\theta_D$) of RcnR at bespoke growth conditions were calculated from transcript abundance of rcnA via Eq. (10):

$$\text{Conditional } \theta_D = 0.99 - 0.98 \times \left(\frac{\text{fold} - \text{change}_{obs} - 1}{\text{fold} - \text{change}_{max} - 1}\right) \qquad (10)$$

where fold-change$_{obs}$ is the observed fold-change in rcnA transcript abundance at the bespoke condition and fold-change$_{max}$ is the maximum fold-change in rcnA transcript abundance at the calibration limit (corresponding to maximum abundance); all fold-changes were determined relative to the defined control condition (untreated LB) corresponding to minimum rcnA transcript abundance (see Supplementary Fig. 27c). Equation (10) defines maximum and minimum transcript abundances as corresponding to $\theta_D$ of 0.01 and 0.99, respectively (see Fig. 8a), and assumes a linear relationship between change in $\theta_D$ and change in transcript abundance.

The intracellular available $[Co^{II}]$ concentration corresponding to each RcnR $\theta_D$ was calculated using known metal affinity, DNA affinities, protein abundance, number of DNA binding sites determined for Salmonella RcnR[9], to calculate the $Co^{II}$-dependent response of E. coli RcnR (93% sequence identity) using excel spreadsheet (Supplementary Dataset 1) and MATLAB code (Supplementary Note 3) available in ref. [9]. The intracellular available $\Delta G_{Co(II)}$ for each condition was calculated using Eq. (11), where $[Co^{II}]$ is the intracellular available $Co^{II}$ concentration, R (gas constant) = $8.314 \times 10^{-3}$ kJ $K^{-1}$ $mol^{-1}$ and T (temperature)

= 298.15 K (see Supplementary Note 1).

$$\text{Intracellular available } \Delta G_{Co(II)} = RT \ln\left[Co^{II}\right] \quad (11)$$

**Estimation of intracellular available $\Delta G_{Zn(II)}$ in LB media**. Fractional responses ($\theta_D$) of Zur and ZntR in LB media were calculated from transcript abundance of *znuA* and *zntA*, via Eqs. (10) and (12), respectively:

$$\text{Conditional } \theta_D = 0.01 + 0.98 \times \left(\frac{\text{fold} - \text{change}_{obs} - 1}{\text{fold} - \text{change}_{max} - 1}\right) \quad (12)$$

where fold-change$_{obs}$ is the observed fold-change in transcript abundance in LB and fold-change$_{max}$ is the maximum fold-change in transcript abundance at the calibration limit (corresponding to maximum abundance); all fold-changes were determined relative to defined control conditions corresponding to minimum transcript abundance (see Supplementary Fig 28a, b). Equation (12) defines maximum and minimum transcript abundances as corresponding to $\theta_D$ of 0.99 and 0.01, respectively, and assumes a linear relationship between change in $\theta_D$ and change in transcript abundance.

The intracellular available [Zn$^{II}$] concentration corresponding to each $\theta_D$ was calculated using known metal affinities, DNA affinities, protein abundance, number of DNA binding sites determined for *Salmonella* homologues, to calculate the Zn$^{II}$-dependent responses of *E. coli* ZntR and Zur (both >92% sequence identity to *Salmonella*) using excel spreadsheet (Supplementary Dataset 1) and MATLAB code (Supplementary Note 3) available in ref. [9]. The intracellular available $\Delta G_{Zn(II)}$ was calculated using Eq. (13), where [Zn$^{II}$] is the intracellular available Zn$^{II}$ concentration, $R$ (gas constant) = $8.314 \times 10^{-3}$ kJ K$^{-1}$ mol$^{-1}$ and $T$ (temperature) = 298.15 K (see Supplementary Note 1).

$$\text{Intracellular available } \Delta G_{Zn(II)} = RT \ln\left[Zn^{II}\right] \quad (13)$$

**Quantification of vitamin B$_{12}$ in *E. coli*\* cultures**. Aliquots (20 mL) of *E. coli*\* culture from each growth condition were taken, and cell pellets frozen at −20 °C. To quantify corrin production (assumed to be predominantly B$_{12}$, since *E. coli*\* contains genes for the complete pathway), *E. coli*\* pellets were thawed, resuspended in H$_2$O (0.2 mL), boiled for 15 min (95 °C) and centrifuged to remove cell debris. An aliquot (10 μL) of each supernatant was applied to *Salmonella typhimurium* AR2680 (Δ*metE*, Δ*cbiB*) pre-inoculated bioassay plates[65], and incubated at 37 °C overnight. Plates were imaged together with a 1-cm$^2$ reference area on black background (see example in Supplementary Data 2) using a Gel-Doc XR + gel documentation system (BioRad; ImageLab software). Images were analysed in MATLAB using the code in Supplementary Software 1 to determine the growth area (in cm$^2$) of each sample. A calibration curve relating growth areas to B$_{12}$ concentration was generated using B$_{12}$ standards (cyanocobalamin; 1–100 nM; quantified by $A_{360\text{ nm}}$ = 27,500 M$^{-1}$ cm$^{-1}$ at pH 10 (ref. [66])) in parallel with *E. coli*\* lysates, using the same batch of bioassay plates (Supplementary Fig. 29a, b). To determine the number of cells in each sample, solutions of *E. coli*\* at varying cell densities (OD$_{600\text{ nm}}$ = 0.2–0.9) were prepared, serially diluted (2000-fold), and the number of cells per mL quantified using a CASY® cell counter. The resulting correlation factor ($4.4 \pm 0.1 \times 10^8$ cells mL$^{-1}$ OD$_{600\text{ nm}}^{-1}$) was used to convert OD$_{600\text{ nm}}$ to cell number (Supplementary Fig. 29c, d).

**Metal content of *E. coli*\* cells**. Aliquots (20 mL) of *E. coli*\* culture from each growth condition were taken and pellets were washed twice with 0.5 M sorbitol, 200 μM EDTA, 20 mM Tris pH 8.5. Cell pellets were suspended in ultrapure 65% (v/v) HNO$_3$ (0.4 mL) to digest (>24 h), then diluted tenfold in 2.5% HNO$_3$ before metal analysis by ICP-MS.

**Statistics and reproducibility**. Sample sizes were chosen based on prior experimental experience, and to give consistent results, following convention in the literature for equivalent analyses. Experiments designed to derive quantitative values used to model or test calculations of metalation were performed in triplicate or more ($n = 3–5$) to enable calculation of SD (listed in Tables or shown as error bars in figures). The number of independent experiments or biologically independent samples is shown in figure legends or footnotes of Tables.

**Reporting summary**. Further information on research design is available in the Nature Research Reporting Summary linked to this article.

## Data availability

All data are available within the article, its Supplementary Information files, plus PDB entry 1NIJ. Source data are provided with this paper.

## Code availability

Equation derivations, Excel spreadsheet (with instructions) constituting a metalation calculator, MATLAB code (with instructions) for use in B$_{12}$ assays and Dynafit scripts are available in Supplementary Note 1, Supplementary Data 1, Supplementary Software 1 and Supplementary Software 2, respectively.

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

## Acknowledgements

This work was supported by a COFUND European Union/Durham University Junior Research Fellowship under EU grant agreement 609412 (T.R.Y.), a Royal Commission for the Exhibition of 1851 Research Fellowship (T.R.Y.), an UKRI Future Leaders Fellowship MR/T019891/1 (R.J.M.), a US-UK Fulbright Commission award (A.G.), Biotechnology and Biological Sciences Research Council awards BB/S009787/1, BB/J017787/1, BB/R002118/1, BB/S002197/1, BB/S014020/1 and Royal Society award INF\R2\180062. We thank Peter Chivers (Durham University, UK) for constructive scientific discussions.

## Author contributions

T.R.Y. conducted the in vitro metal-binding experiments for CobW, GTP-hydrolysis assays, in vivo gene expression experiments and B$_{12}$-production experiments. A.W.F. conducted the in vitro metal-binding experiments for YjiA. A.G. conducted the in vitro metal-binding experiments for YeiR. T.R.Y. and M.A.M. developed the experimental protocols for determining metal sensor responses by qPCR. M.A.M. derived equations for the metalation calculator and produced the spreadsheet. R.J.M. and D.O. generated the MATLAB code for analysis of B$_{12}$ bioassays. E.D. generated the CobW expression plasmid. E.D. and M.J.W. donated the B$_{12}$-producing *E. coli*\* strains and advised on B$_{12}$ biochemistry. E.D., M.J.W. and T.R.Y. co-designed the B$_{12}$-production experiments. T.R.Y. and N.J.R. drafted the manuscript with input from A.W.F. T.R.Y. and N.J.R., in conjunction with A.W.F., A.G., M.A.M. and D.O., interpreted the significance of the data. T.R.Y. and N.J.R. had overall responsibility for the design and management of the project. All authors reviewed the results and edited and approved the final version of the manuscript.

## Competing interests

The authors declare no competing interests.
