## [Peer Review File · Nature Communications]

REVIEWER COMMENTS

Reviewer #1 (Remarks to the Author):

Young et al report a framework to predict the metal occupancy of metalloenzymes under in vivo conditions, and apply this process, combined with rigorous in vitro characterization, and link cobalt occupancy of *Rhodobacter capsulatus* CobW with cobalamin biosynthesis when expressed in engineered *E. coli* cells. Previous studies implicated CobW involvement in cobalamin biosynthesis, but have not yet demonstrated how the protein selects for Cobalt over other intracellular metals. The bulk of the manuscript involves a large body of biochemical characterization of the purified CobW protein, demonstrating that like other G3E GTP hydrolases, nucleotide binding is required to activate binding of the cognate metal. Further characterization reveals that the protein preferentially binds Zinc under in vitro conditions to Cobalt. This apparent paradox is resolved by comparing the intracellular availability of these metals (as expressed by standard free energies) to the intrinsic biochemical preference, demonstrating an in vivo preference for cobalt.

The work is an effective continuation of previous work by the Robinson group in understanding protein metallation, and impactful because, in combination with these previous reports, it demonstrates that their bacterial metal sensors have broad general use. Overall this is high quality work communicated in a well-written paper, and I have only minor comments.

1) I do not have a good sense as to how the engineered B12-producing *E. coli* strain was constructed, and the literature reference is unclear. Is the strain from previous literature, and, if so, can it be more clearly cited? If not, a procedure for its construction should be included.

2) A key theoretical underpinning of the paper is that CobW metallation is the step that leads to selective Cobalt insertion over Zinc (or other metals). Under the 1 and 3 μM Co levels, the predicted occupancy of CobW suggests that Zincobalamin may be produced under these conditions. Given the different chemical properties between the two metals, I do not expect that Zincobalamin would be detected using the bioassay strains used. While explicit experimental data addressing this hypothesis are difficult, a sentence in the results section stating that detection of zincobalamin was not able to be performed would be appreciated.

3) Please include other replicates from figures 1e, 1h, and 3e in the supplemental data.

4) Is the data in figures 1d, 1g, 2a, 2b, and 5b-d single replicates?

5) Page 3, line 23 — B12 is also not made or required by fungi.

Page 3, line 24-25 — B12 is also found in other animal products, esp. meat, so it might be better to mention this instead of just its presence/absence in cow's milk.

6) Please explain briefly the competition assay the first time it is presented (page 6, line ~4), and the full chemical name and function of *fura2* (page 6, line 18) for a broader audience.

7) Page 6, line 13: rephrase to "CoII binds 1000-fold more tightly to CobW with GTP than GDP"

Reviewer #2 (Remarks to the Author):

In their manuscript, 'Calculating metalation in cells reveals CobW acquire Co(II) for vitamin B12 biosynthesis upon binding nucleotides,' Robinson and co-workers address the question of how proteins acquire the correct metal ion to function. This fundamental question is not straightforward – using metal binding affinities alone as the predictor – proteins often will be paired with the 'wrong' metal – and, as Robinson has shown us over the years - affinities are not sufficient to predict metalation. Here the focus is on CobW, a protein associated with aerobic biosynthesis of cobalamin (vitamin B12) and thought to be a cobalt co-factored protein. Experiments to test the hypothesis that CobW acquires cobalt for delivery to the B12 biosynthetic pathway are described. The experiments use an engineered *E. coli* strain (native *E. coli* does not produce B12, but has the functional pathways), and the engineered *E. coli* strain is tractable and the data obtained can be

evaluated in the context of metals in metalloregulatory proteins in *E. coli* (which Robinson has extensively studied). The major finding is that CobW is indeed a Co-cofactored protein, but requires Mg and GTP association for Co binding. To make this finding, Robinson and co-workers systematically measure the affinity of multiple first row transition metals for CobW in the presence and absence of Mg and GTP, using an approach which incorporates the use of chemical potentials (ΔG) and the assumption of an idealized cell for metal occupancy (based upon extensive work performed by this laboratory on *Salmonella*). This is a clearly presented, and deftly carried out body of work that both identifies the likely metal co-factor for CobW, but also further highlights the complexity of metalation of proteins in the cellular milieu. Important findings include the presence of a tetrahedral Co(II) center (via UV-vis), that presumably utilizes two conserved cysteine residues in the CobW protein along with (perhaps) ligands from GTP as well as a link to vitamin B biosynthesis in cells.

The authors also provide a nice "metallation calculator" – which is an excel spreadsheet pre-programmed to calculate both ΔG and intracellular ΔG to calculate the metalation state of a protein of interest. This will provide an important tool to researchers who are interested in determining metalation state of a new protein of interest. Overall, this is an important body of work, and I recommend publication.

Questions to be addressed:

- (1) Although the excel version of the calculator is user-friendly (I tried it!), I wonder if a web version could be made to provide more access (rather than digging in the supplemental and downloading the calculator)?
- (2) Can the authors speculate more on the potential ligands for Co in CobW. Other than the conserved two cysteine residues, what other ligands do the authors think would make up the tetrahedral site, and could ligand identity also play a role in the modulating metalation?
- (3) In vivo link – I appreciated that the rigorous thermodynamic measurements were linked to an 'in vivo' measure – vitamin B12 biosynthesis – but I wondered if the authors also measured the total metal content of the cells (or could this be done?), so that the values for vitamin B12 biosynthesis and cell number could be connected to metal content?

Reviewer #3 (Remarks to the Author):

The manuscript by Young et al. seeks to characterize the metal binding properties, namely the preferred metal cofactor, in the putative cobalt chaperone, CobW. Two questions are addressed: first, how are metal binding properties affected by the presence of the cognate nucleotide, GTP? Second, given the context of the authors' previous work characterizing intracellular metal availability in terms of free energies, what is the predicted identity of the metal cofactor in vivo? To address the latter, the authors developed a "metallation-calculator", that is, a spreadsheet that allows for the calculation of the metalation state of a molecule of in terms of the %occupancy of each d-block metal.

To this end, the first key finding is that the affinity of Co(II) for the protein is enhanced by the presence of Mg(II)-GTP, to a significant degree over Mg(II)-GDP, suggesting an allosteric role of the cofactor for tight metal binding.

Having determined the nucleoside-dependence on binding, the authors calculated the binding affinities of various d-block metal ions to CobW in the presence of Mg(II)-GTP and utilized this information to determine, using their previously established free energy-based calculations, which ions can be expected to bind with reasonable probability inside the cell. It was determined that both Zn(II) and Co(II) could compete for binding. While Zn(II) has the higher affinity under unsupplemented environments, the calculator predicts that increasing [Co(II)] in solution would allow for increased binding of Co(II) from 10 to 97% at 0 - 30 μM Co(II) in the media. The calculations were closely correlated with experimental assessment in *E. coli*.

The work is of interest to the bioinorganic community, as metal speciation remains a challenging question. The calculator is especially useful for broader applications beyond CobW. The presented

work is rigorous and resulting conclusions are fitting in accordance with the data. It is gratifying to see quantitative predictions and analysis that justifies the proposed role of CobW as a chaperone in B12 synthesis. However, this reviewer believes that the key findings and products of this manuscript may be too narrow in scope for this journal in its current state and may be more appropriate for a more specialized journal. Additionally, while the calculator is an exciting tool, this reviewer recommends additional demonstration of applicability/accuracy of the calculator to facilitate its utility for other researchers.

Regarding the scope: while the presented work provides quantitative justification for the interaction of CobW with Co(II), the idea that this binding occurs and that the protein plays a role in B12 synthesis is not novel. It is not clear to this reviewer from the way the manuscript is written what makes this finding important to the wider field, and perhaps this can be fixed by a rewriting of the introduction or discussion as to the degree the quantitative finding might impact not only the study of CobW but perhaps a larger set of proteins.

I do think that novelty is in the approach the authors take to validate the hypothesis, particularly in predicting metal binding and the development of the calculator when there is possible competition between two metals. However, at its current state, only applying the calculator to a single system seems premature with respect to claiming the broader applicability of the tool. This particular issue may be addressed by expanding the calculator and correlated experimental validation to other pairs of metals, to mutant forms of the protein that affect affinities and potentially shift the competition, or to other proteins that are perhaps more controversial in their role/metal occupancy.

I want to summarize that the work presented is rigorous and appreciated this reviewer and should be published, but perhaps elsewhere.

A minor comment: the writing itself seems to be geared towards a specialized audience. Understanding the larger context was at times challenging. One suggestion that can address this is to devote more sentences in the introduction, discussion, or both to explain the problem of predicting metal speciation in the context beyond the system that the authors are working with.

Point-by-point responses to reviewers' comments.

We thank all three reviewers for encouraging and advising us to produce a substantially improved manuscript.

REVIEWER COMMENTS

Reviewer #1 (Remarks to the Author):

Young et al report a framework to predict the metal occupancy of metalloenzymes under in vivo conditions, and apply this process, combined with rigorous in vitro characterization, and link cobalt occupancy of *Rhodobacter capsulatus* CobW with cobalamin biosynthesis when expressed in engineered *E. coli* cells. Previous studies implicated CobW involvement in cobalamin biosynthesis, but have not yet demonstrated how the protein selects for Cobalt over other intracellular metals. The bulk of the manuscript involves a large body of biochemical characterization of the purified CobW protein, demonstrating that like other G3E GTP hydrolases, nucleotide binding is required to activate binding of the cognate metal. Further characterization reveals that the protein preferentially binds Zinc under in vitro conditions to Cobalt. This apparent paradox is resolved by comparing the intracellular availability of these metals (as expressed by standard free energies) to the intrinsic biochemical preference, demonstrating an in vivo preference for cobalt.

The work is an effective continuation of previous work by the Robinson group in understanding protein metallation, and impactful because, in combination with these previous reports, it demonstrates that their bacterial metal sensors have broad general use. Overall this is high quality work communicated in a well-written paper, and I have only minor comments.

1) I do not have a good sense as to how the engineered B12-producing *E. coli* strain was constructed, and the literature reference is unclear. Is the strain from previous literature, and, if so, can it be more clearly cited? If not, a procedure for its construction should be included.

Both plasmid-based and chromosomally-integrated *E. coli* strains, up to the point of cobalt insertion, have been described previously (Deery et al, *Nat. Chem. Biol.*, 2012, 8: 933-940; Kieninger et al, *Angew. Chem. Int.*, 2019, 31: 10756-10760). Here, additional genes have been added to complete the pathway. Additional details of these strains have now been added to the methods (pg 24, line 13) as requested.

2) A key theoretical underpinning of the paper is that CobW metallation is the step that leads to selective Cobalt insertion over Zinc (or other metals). Under the 1 and 3 μM Co levels, the predicted occupancy of CobW suggests that Zincobalamin may be produced under these conditions. Given the different chemical properties between the two metals, I do not expect that Zincobalamin would be detected using the bioassay strains used. While explicit experimental data addressing this hypothesis are difficult, a sentence in the results section stating that detection of zincobalamin was not able to be performed would be appreciated.

The reviewer is correct that the B₁₂ assay will only detect functional (Co^{II}-containing) corrins. However, this may include intermediates after the Co^{II} insertion step and a note to this effect has been added to the results text. The CobNST chelatase introduces Co^{II} into ring-contracted corrins, in the late-insertion pathway, and *in vitro* studies do not observe insertion of Zn^{II} at this step (Debussche et al, *J. Bacteriology*, 1992, 174: 7445-7451), however Zn^{II} has the potential to act as a competitive inhibitor. Again, a note to this effect has been added to the results text, in line with the

suggestion from the referee: 'Corrin concentrations (presumed to be predominantly B₁₂, noting that intermediates after Co^{II} insertion may also be detected, and that Zn^{II} may competitively inhibit the chelatase complex but not insert into ring-contracted corrins³⁶) were measured in *E. coli*''

3) Please include other replicates from figures 1e, 1h, and 3e in the supplemental data.

Replicate data for Figs. 1e,h have been provided in Supplementary Figure 1c,e.

Replicate data for Fig. 3e (now Fig. 2g) is in Supplementary Fig. 6 - a reference has been added to the legend of Fig. 3 (now Fig. 2) for clarity.

4) Is the data in figures 1d, 1g, 2a, 2b, and 5b-d single replicates?

The legends of Figs. 1d, 1g, 2 and 5b-d have been updated to specify n explicitly, in each case.

Replicate data for Figs. 1d,g and 2b (n=2) have been deposited in Supplementary Figs. 1b,d and 3e.

Replicate data for Fig. 2a (n=3 independent experiments using different competitors and/or concentrations) are presented in Supplementary Figs. 3c,d (now added) and 4a,b (as originally presented, but now referenced in legend of Fig. 2 for clarity).

Data in Fig. 5b-d (now Fig. 4b-d) are each single replicates (comprising a total of n=4 independent experiments to determine Zn^{II} affinity), now stated in the figure legend.

5) Page 3, line 23 — B12 is also not made or required by fungi.

Page 3, line 24-25 — B12 is also found in other animal products, esp. meat, so it might be better to mention this instead of just its presence/absence in cow's milk.

The text has been updated and now states that 'Vitamin B₁₂ is an essential nutrient for human health but is neither made nor required by plants or fungi' and that 'meat and dairy products provide a dietary source'.

6) Please explain briefly the competition assay the first time it is presented (page 6, line ~4), and the full chemical name and function of fura2 (page 6, line 18) for a broader audience.

The following explanation of competition assays has been added to page 6:

'Due to the tight coordination of Co^{II} to nucleotide-bound forms of CobW (ie no measurable dissociation at the micromolar-range protein concentrations required for detection), it was necessary to employ competition assays, whereby Co^{II} is partitioned between the protein and a ligand of well-matched and defined Co^{II} affinity, for reliable quantification of metal-binding affinities'

The chemical name for fura-2 '5-Oxazolecarboxylic acid, 2-(6-(bis(carboxymethyl)amino)-5-(2-(2-(bis(carboxymethyl)amino)-5-methylphenoxy)ethoxy)-2-benzofuranyl)-pentapotassium salt' has been added to the methods section (under 'Determination of CobW metal affinities via ligand competition') as requested by the reviewer. We agree with the reviewer that it is valuable for its function to be defined in the main text, and the sentence now reads:

'Co^{II} affinities of CobW and Mg^{II}GDP-CobW were determined via competition with the probe ligand fura-2 (Fig. 2c,d, Supplementary Fig. 4f-I and Supplementary Tables 1,2), which undergoes fluorescence quenching upon Co^{II}-binding'

7) Page 6, line 13: rephrase to "Coll binds 1000-fold more tightly to CobW with GTP than GDP"

This statement has now been rephrased in the text, as suggested.

Reviewer #2 (Remarks to the Author):

In their manuscript, 'Calculating metalation in cells reveals CobW acquire Co(II) for vitamin B12 biosynthesis upon binding nucleotides,' Robinson and co-workers address the question of how proteins acquire the correct metal ion to function. This fundamental question is not straightforward – using metal binding affinities alone as the predictor – proteins often will be paired with the 'wrong' metal – and, as Robinson has shown us over the years - affinities are not sufficient to predict metalation. Here the focus is on CobW, a protein associated with aerobic biosynthesis of cobalamin (vitamin B12) and thought to be a cobalt co-factored protein. Experiments to test the hypothesis that CobW acquires cobalt for delivery to the B12 biosynthetic pathway are described. The experiments use an engineered E. coli strain (native E. coli does not produce B12, but has the functional pathways), and the engineered E. coli strain is tractable and the data obtained can be evaluated in the context of metals in metalloregulatory proteins in E. coli (which Robinson has extensively studies). The major finding is that CobW is indeed a Co-cofactored protein, but requires Mg and GTP association for Co binding. To make this finding, Robinson and co-workers systematically measure the affinity of multiple first row transition metals for CobW in the presence and absence of Mg and GTP, using an approach which incorporates the use of chemical potentials (ΔG) and the assumption of an idealized cell for metal occupancy (based upon extensive work performed by this laboratory on Salmonella). This is a clearly presented, and deftly carried out body of work that both identifies the likely metal co-factor for CobW, but also further highlights the complexity of metalation of proteins in the cellular milieu. Important findings include the presence of a tetrahedral Co(II) center (via UV-vis), that presumably utilizes two conserved cysteine residues in the CobW protein along with (perhaps) ligands from GTP as well as a link to vitamin B biosynthesis in cells.

The authors also provide a nice "metallation calculator" – which is an excel spreadsheet pre-programmed to calculate both ΔG and intracellular ΔG to calculate the metalation state of a protein of interest. This will provide an important tool to researchers who are interested in determining metalation state of a new protein of interest. Overall, this is an important body of work, and I recommend publication.

Questions to be addressed:

(1) Although the excel version of the calculator is user-friendly (I tried it!), I wonder if a web version could be made to provide more access (rather than digging in the supplemental and downloading the calculator)?

The authors are very enthusiastic about this reviewers' suggestion to make the calculator available on the web. However, we are keen that this manuscript will become the source reference for the calculator and therefore the web-based version should follow after publication. Web-developers have been recruited to create this.

(2) Can the authors speculate more on the potential ligands for Co in CobW. Other than the conserved two cysteine residues, what other ligands do the authors think would make up the tetrahedral site, and could ligand identity also play a role in the modulating metalation?

The 3 conserved cysteines, CxCC, in CobW homologues are noted as potential ligands and these are now highlighted in an additional supplementary figure (Supplementary Fig. 29). Supplementary Figure 29 also identifies a further pair of cysteines which are conserved in *bona-fide* CobW proteins but are missing from homologues YeiR, YjiA and other putative Zn^{II}-GTPases. In response to comments of reviewer 3, data on Mg^{II}GTP-YeiR and Mg^{II}GTP-YjiA have now been added to the manuscript, revealing that these proteins will preferentially bind Zn^{II} *in vivo*, not because they have

tighter Zn^{II} affinities than Mg^{II}GTP-CobW but because they have weaker Co^{II} affinities. In response to comment 2 of reviewer 2 we have therefore also added a note in the discussion that the additional pair of Cys in CobW (now highlighted on Supplementary Fig. 29) may also play a role in modulating metalation in favour of Co^{II}: *'there is now a quest to understand why Co^{II} affinities are weaker for Mg^{II}GTP-bound YeiR and YjiA (and, hypothetically, ZigA and ZagA), which bind predominantly to Zn^{II} as a result (Fig. 6). Notably a further pair of conserved Cys residues (C₅₆, C₆₁) in CobW's are absent from the homologues (Supplementary Fig. 29).'*

(3) In vivo link – I appreciated that the rigorous thermodynamic measurements were linked to an 'in vivo' measure – vitamin B12 biosynthesis – but I wondered if the authors also measured the total metal content of the cells (or could this be done?), so that the values for vitamin B12 biosynthesis and cell number could be connected to metal content?

These data have now been collected, included in an additional table (Supplementary Table 9) and described in the results: *'As anticipated, total cellular cobalt increases with supplementation, and the amount of cobalt in B₁₂ is <10% of the total cellular cobalt (Supplementary Table 9). The number of additional atoms accumulated per cell exceeds the amount predicted if Co^{II} were not buffered, noting that the internal buffered concentration at 10 μM exogenous Co^{II} is 1.9 nM (Figure 8, Supplementary Table 8), and that only 1 atom per cell volume (approximately 1 femtolitre) equates to 1.7 nM.'*

Reviewer #3 (Remarks to the Author):

The manuscript by Young et al. seeks to characterize the metal binding properties, namely the preferred metal cofactor, in the putative cobalt chaperone, CobW. Two questions are addressed: first, how are metal binding properties affected by the presence of the cognate nucleotide, GTP? Second, given the context of the authors' previous work characterizing intracellular metal availability in terms of free energies, what is the predicted identity of the metal cofactor in vivo? To address the latter, the authors developed a "metalation-calculator", that is, a spreadsheet that allows for the calculation of the metalation state of a molecule of in terms of the %occupancy of each d-block metal.

To this end, the first key finding is that the affinity of Co(II) for the protein is enhanced by the presence of Mg(II)-GTP, to a significant degree over Mg(II)-GDP, suggesting an allosteric role of the cofactor for tight metal binding.

Having determined the nucleoside-dependence on binding, the authors calculated the binding affinities of various d-block metal ions to CobW in the presence of Mg(II)-GTP and utilized this information to determine, using their previously established free energy-based calculations, which ions can be expected to bind with reasonable probability inside the cell. It was determined that both Zn(II) and Co(II) could compete for binding. While Zn(II) has the higher affinity under unsupplemented environments, the calculator predicts that increasing [Co(II)] in solution would allow for increased binding of Co(II) from 10 to 97% at 0 - 30 μM Co(II) in the media. The calculations were closely correlated with experimental assessment in *E. coli*.

The work is of interest to the bioinorganic community, as metal speciation remains a challenging question. The calculator is especially useful for broader applications beyond CobW. The presented work is rigorous and resulting conclusions are fitting in accordance with the data. It is gratifying to see quantitative predictions and analysis that justifies the proposed role of CobW as a chaperone in B12 synthesis. However, this reviewer believes that the key findings and products of this manuscript may be too narrow in scope for this journal in its current state and may be more appropriate for a more specialized journal. Additionally, while the calculator is an exciting tool, this reviewer recommends additional demonstration of applicability/accuracy of the calculator to facilitate its utility for other

researchers.

Regarding the scope: while the presented work provides quantitative justification for the interaction of CobW with Co(II), the idea that this binding occurs and that the protein plays a role in B12 synthesis is not novel. It is not clear to this reviewer from the way the manuscript is written what makes this finding important to the wider field, and perhaps this can be fixed by a rewriting of the introduction or discussion as to the degree the quantitative finding might impact not only the study of CobW but perhaps a larger set of proteins.

The manuscript has been substantially broadened (beyond experimentally establishing the role of CobW and its mechanism of cobalt acquisition), in part by including new data for additional (*Salmonella*) proteins, YeiR and YjiA, that demonstrate the wider applicability/accuracy of the calculator and its utility for other researchers (see response to later comments).

Parts of the manuscript have also been extensively rewritten and reordered to clarify the importance of the work to the wider field and this is also reflected in a revised title and abstract (the extent of these changes is evident from the yellow highlights and margin notes on the revised manuscript, and also see response to later comments).

We thank the reviewer for encouraging us to make these major changes, which have created a paper that should now be appreciated by a very broad audience. The revised manuscript concludes *'with almost a half of enzymes requiring metals, an ability to calculate metalation in vivo should have broad applicability in optimising (or subverting) metalation in biotechnology. The calculator (Supplementary Data 1) can be widely used to understand metalation and mismetalation of proteins that acquire Mg^{II}, Mn^{II}, Fe^{II}, Co^{II}, Ni^{II}, Cu^I or Zn^{II} from the milieu inside living cells.'*

I do think that novelty is in the approach the authors take to validate the hypothesis, particularly in predicting metal binding and the development of the calculator when there is possible competition between two metals. However, at its current state, only applying the calculator to a single system seems premature with respect to claiming the broader applicability of the tool. This particular issue may be addressed by expanding the calculator and correlated experimental validation to other pairs of metals, to mutant forms of the protein that affect affinities and potentially shift the competition, or to other proteins that are perhaps more controversial in their role/metal occupancy.

The use of the calculator has been expanded to other proteins (YeiR and YjiA) that are more controversial in their roles and metal occupancies. These proteins are related to CobW and, due to their uncertain role in handling zinc, other members of the research group had already collected these data and therefore there are two additional authors (Foster and Glasfeld) on the revised manuscript. The outcome supports the putative roles of these G3E GTPases in handling zinc.

Three new results sections *'Related GE3 GTPase YeiR prefers Zn^{II} in idealised cells'*, *'Related GE3 GTPase YjiA prefers Zn^{II} in idealised cells'* and *'Mg^{II}GTP-CobW outcompetes Mg^{II}GTP-YeiR for Co^{II}'* have been added. Two new figures (Fig. 6 and Fig. 7), additional data in Table 1, 10 new supplementary figures (Supplementary Figs. 15-24) and 3 new supplementary tables (Supplementary Tables 5-7) are new results for YeiR and YjiA. 15 new Dyanfit scripts, used for fitting the new data for YeiR and YjiA, have been added to Supplementary Note 3. To accommodate this broader scope, Figures 2 and 3 have been combined and Figure 9 has been moved to the supplements (Supplementary Figure 28).

These extra data provide further novelty in discovering that a weakened cobalt affinity can generate a zinc protein inside a cell: This is summarised in the discussion, *'In contrast, for nucleotide-bound forms of YeiR and YjiA $\Delta\Delta G$ for Zn^{II} is more negative than $\Delta\Delta G$ for Co^{II} making these deduced Zn^{II} proteins'* and *'Zn^{II} is identified as the preferred metal for nucleotide bound forms of YeiR and YjiA: This is due to their weaker affinities for Co^{II} relative to Mg^{II}GTP-CobW (Fig. 7, Supplementary Table*

5), rather than tighter affinities for Zn^{II}' and highlighted in the revised abstract '*The calculator also reveals that related GTPases with comparable Zn^{II} affinities to CobW, preferentially acquire Zn^{II} due to their relatively weaker Co^{II} affinities*'.

This discovery further highlights the importance of accounting for competition between metals in predicting metal-binding inside cells.

A minor comment: the writing itself seems to be geared towards a specialized audience. Understanding the larger context was at times challenging. One suggestion that can address this is to devote more sentences in the introduction, discussion, or both to explain the problem of predicting metal speciation in the context beyond the system that the authors are working with.

The introduction has been restructured, redrafted, and more sentences added, and begins with the problem of predicting metal speciation inside a cell. Extraneous material has been removed to retain this focus. The final paragraph of the introduction starts with the statement '*The purpose of this work was to make it widely possible to quantify metal occupancy of proteins and other molecules in vivo based on thermodynamic parameters. The cognate metals of proteins can thus be identified where this was uncertain, and the contributions of additional mechanisms that enable metalation (such as molecular interactions or bespoke growth conditions) exposed*'.

The discussion has similarly been rewritten to focus on the problem of predicting metal speciation and to more clearly explain the larger context. The discussion begins '*Here we relate metal affinities of three putative metallochaperones to a thermodynamic framework, identifying their cognate metals which align with previous speculations^{16,20,25} (Figs. 5, 6 and Table 1)*'. It goes on to state '*these data illustrate the value of using the metalation calculator provided as Supplementary Data 1, which can now be broadly applied to metal-speciation in the context of intracellular competition*', '*the previously intractable challenge to understand inter-metal competition in a cell now becomes tractable*' and '*by monitoring the responses of sensors for different metals it will be possible to define available ΔG , and predict protein occupancies with diverse metals, in different growth conditions*'. The broader context is reflected in the concluding paragraph, as noted earlier (in response to comment 1 of reviewer 3).

As above, we appreciate being encouraged to redraft the manuscript for a broad audience and to emphasise the larger context/implications/applications of these discoveries.

We hope that you will enjoy reading the revised version of this manuscript.

Yours sincerely,

Tessa Young and Nigel Robinson

REVIEWER COMMENTS

Reviewer #1 (Remarks to the Author):

The authors addressed all of the concerns raised in the first review, and as a consequence the manuscript is substantially improved and is written significantly more clearly. I agree with Reviewer 3 that broader applicability would strengthen the manuscript, and believe that by demonstrating this in two other metal-binding proteins in vitro leading to an in vivo prediction, the authors have gone far to address the broader applicability of the system. Would it be possible to validate one of these in vivo predictions (for example, using ICP-MS of at least one of the two purified proteins)?

One concern that arises is that calculated in vivo metal occupancies are for Salmonella, since comparable in vivo measurements have not been made for other bacteria. It would help to state explicitly that this calculator predicts in vivo metal occupancy best in that context (and for closely related species). Key areas where this could be addressed in the text include:

- page 3, line 33: We determine metal affinities of CobW, YeiR and YjiA, and calculate their expected in vivo metal occupancies in Salmonella.
- When introducing the calculator on page 9.
- The subsection titles (page 9, line 31 and page 10, line 22): "in idealised Salmonella cells."

Reviewer #2 (Remarks to the Author):

The authors have responded to my questions satisfactorily. Even though I didn't suggest this change, I appreciate that the authors expanded to include additional proteins (in response to one of the other referees). This strengthens the work. I recommend publication without any additional edits.

Reviewer #3 (Remarks to the Author):

My initial concern with the manuscript was not one of scientific rigor and validity, but rather of scope, given the audience of Nat. Comm. The authors have diligently addressed this concern, providing applications of the calculator to more controversial systems. In this reviewer's opinion, this manuscript is much-improved and its current form makes its impact for the broader bioinorganic community evident. I think this work will be a much-needed contribution to the field. For these reasons, and for the diligence of the authors in the revisions, I am pleased to recommend publication of the manuscript in its current state.

Re: Point-by-point responses to reviewers' comments.

The revised manuscript addresses all of the points raised by Reviewer 1. Items in the main text which have been changed are highlighted yellow.

Reviewer #1 (Remarks to the Author):

The authors addressed all of the concerns raised in the first review, and as a consequence the manuscript is substantially improved and is written significantly more clearly. I agree with Reviewer 3 that broader applicability would strengthen the manuscript, and believe that by demonstrating this in two other metal-binding proteins *in vitro* leading to an *in vivo* prediction, the authors have gone far to address the broader applicability of the system. Would it be possible to validate one of these *in vivo* predictions (for example, using ICP-MS of at least one of the two purified proteins)?

Metals are often lost or exchanged during purification procedures making it notoriously challenging to determine *in vivo* metal occupancies post-extraction (hence the need for a metalation calculator). Nonetheless we attempted the requested experiment and some Zn^{II} was detected in association with YeiR. These data are included as Supplementary Figures 22 and 23, and the legend to Supplementary Figure 22 lists the caveats associated with the interpretation of such data.

These new data are referred to on page 11, line 27 of the main text.

One concern that arises is that calculated *in vivo* metal occupancies are for *Salmonella*, since comparable *in vivo* measurements have not been made for other bacteria. It would help to state explicitly that this calculator predicts *in vivo* metal occupancy best in that context (and for closely related species).

The following statement has been added to the discussion (page 15, line 21): 'The calculator should be most accurate in *Salmonella* and closely-related species such as *E. coli*. However, metal availabilities can also be adjusted (and/or simulated) to account for species differences, noting that the dynamic ranges of available ΔG values might be similar even when total cellular metal changes greatly between species.'

The description of the metalation calculator in Supplementary Data 1 has also been amended to read: 'This spreadsheet allows the calculation of the metalation state of a molecule of interest *in vivo* (anticipated to be most accurate in *Salmonella* Typhimurium and closely related species, and can also be used for simulating hypothetical- affinities or availabilities).'

Key areas where this could be addressed in the text include:

- page 3, line 33: We determine metal affinities of CobW, YeiR and YjiA, and calculate their expected *in vivo* metal occupancies in *Salmonella*.

Page 4, line 37 now states: 'We determine metal affinities of CobW, YeiR and YjiA, and calculate their *in vivo* metal occupancies (in *Salmonella* and closely related species)'

- When introducing the calculator on page 9.

Page 10, line 22 now states: 'Thus, we developed a metalation calculator (based on *Salmonella*, Supplementary Data 1)'

- The subsection titles (page 9, line 31 and page 10, line 22): "in idealised *Salmonella* cells."

These two subsection titles now read 'in idealised *Salmonella*'. Additionally, the preceding subsection title now reads: 'Mg^{II}GTP-CobW selects Co^{II} in idealised (*Salmonella*) cells'.

Reviewer #2 (Remarks to the Author):

The authors have responded to my questions satisfactorily. Even though I didn't suggest this change, I appreciate that the authors expanded to include additional proteins (in response to one of the other referees). This strengthens the work. I recommend publication without any additional edits.

No further changes required.

Reviewer #3 (Remarks to the Author):

My initial concern with the manuscript was not one of scientific rigor and validity, but rather of scope, given the audience of Nat. Comm. The authors have diligently addressed this concern, providing applications of the calculator to more controversial systems. In this reviewer's opinion, this manuscript is much-improved and its current form makes its impact for the broader bioinorganic community evident. I think this work will be a much-needed contribution to the field. For these reasons, and for the diligence of the authors in the revisions, I am pleased to recommend publication of the manuscript in its current state.

No further changes required.

We hope that you will enjoy reading the revised version of this manuscript.

Yours sincerely,

Tessa Young and Nigel Robinson

REVIEWERS' COMMENTS

Reviewer #1 (Remarks to the Author):

The authors have addressed all concerns raised in the previous version of the manuscript. I have no further concerns and I recommend it for publication.

During the time period after the authors were asked to correct the sentence beginning on line 87 to state that fungi do not make or require B12, a new bioRxiv preprint reported that certain fungi are indeed predicted to produce B12. (<https://doi.org/10.1101/2020.10.13.337048>) The authors may wish to update this sentence once again.

REVIEWERS' COMMENTS

Reviewer #1 (Remarks to the Author):

The authors have addressed all concerns raised in the previous version of the manuscript. I have no further concerns and I recommend it for publication.

During the time period after the authors were asked to correct the sentence beginning on line 87 to state that fungi do not make or require B12, a new bioRxiv preprint reported that certain fungi are indeed predicted to produce B12. (<https://doi.org/10.1101/2020.10.13.337048>) The authors may wish to update this sentence once again.

The reference to fungi has been removed from the text, as requested.

We hope that you are staying safe and well in these challenging times and look forward to hearing more in the near future.

Yours sincerely,

Tessa Young and Nigel Robinson